# Direct reciprocity between individuals that use different strategy spaces

**Laura Schmid** [1]*, **Christian Hilbe** [2], **Krishnendu Chatterjee** [1], **Martin A. Nowak** [3]

**1** IST Austria, Klosterneuburg, Austria, **2** Max Planck Research Group Dynamics of Social Behavior, Max Planck Institute for Evolutionary Biology, Plön, Germany, **3** Department of Mathematics, Department of Organismic and Evolutionary Biology, Harvard University, Cambridge, Massachusetts, United States of America

* lschmid@ist.ac.at

## Abstract

In repeated interactions, players can use strategies that respond to the outcome of previous rounds. Much of the existing literature on direct reciprocity assumes that all competing individuals use the same strategy space. Here, we study both learning and evolutionary dynamics of players that differ in the strategy space they explore. We focus on the infinitely repeated donation game and compare three natural strategy spaces: memory-1 strategies, which consider the last moves of both players, reactive strategies, which respond to the last move of the co-player, and unconditional strategies. These three strategy spaces differ in the memory capacity that is needed. We compute the long term average payoff that is achieved in a pairwise learning process. We find that smaller strategy spaces can dominate larger ones. For weak selection, unconditional players dominate both reactive and memory-1 players. For intermediate selection, reactive players dominate memory-1 players. Only for strong selection and low cost-to-benefit ratio, memory-1 players dominate the others. We observe that the supergame between strategy spaces can be a social dilemma: maximum payoff is achieved if both players explore a larger strategy space, but smaller strategy spaces dominate.

**Data Availability Statement:** All simulations and numerical calculations have been performed with Fortran 90. The code used to calculate payoffs and simulate the tournaments is available online at https://osf.io/se863/?view_only= 7ba5d63ca5d6402c8c081b2fdcd682be! The

## Author summary

Direct reciprocity can lead to cooperation between individuals who meet in repeated encounters. The shadow of the future casts an incentive to cooperate. If I cooperate today, you may cooperate tomorrow. But if I defect today, you may defect tomorrow. In most studies of direct reciprocity it is assumed that both players explore the same space of possible strategies. In contrast, here we study interactions between players that use different strategy spaces and therefore utilize different memory capacities. Surprisingly, we find that more complex strategy spaces often lose out against simpler ones. The social optimum, however, is achieved if all players use the more complex space. Therefore, the game between strategy spaces becomes a higher order social dilemma.

dynamics of the supergame was visualized with Mathematica 11 and Dynamo 3S.

**Funding:** This work was supported by the European Research Council (https://erc.europa.eu/) CoG 863818 (ForM-SMArt) (to K.C.), and the European Research Council Starting Grant 850529: E-DIRECT (to C.H.). The funders had no role in study design, data collection and analysis, decision to publish, or preparation of the manuscript.

**Competing interests:** The authors have declared that no competing interests exist.

## Introduction

Direct reciprocity is a mechanism for the evolution of cooperation [1–3]. It is based on the insight that people have more of an incentive to cooperate when they meet repeatedly [4, 5]. To formalize this concept, researchers study optimal behavior in games like the repeated prisoner's dilemma [6]. Here, individuals repeatedly decide whether to cooperate or defect with their co-player. While each player prefers to defect if they only interact once, cooperation becomes feasible when they interact over multiple rounds [7–9]. Because of its simple structure, the repeated prisoner's dilemma has become a main paradigm for conceptualizing reciprocity [2], and even has applications beyond human behavior [10]. It can explain why humans exchange favors, but also why stickleback fish alternate in predator scouting [11], or why vampire bats share food [12].

In repeated games, players can take into account the previous interactions when deciding what to do next. To make these decisions, player use a strategy, which is a rule that specifies whether or not to cooperate given the history of the game. Often, the set of feasible strategies is constrained by how many past decisions the player is able to remember. The most restrictive assumption is that players do not remember any past events. In that case, they can only use unconditional strategies, such as cooperating with a probability that is independent of previous interactions. Alternatively, players may remember the co-player's last move, which allows them to use reactive strategies [13]. Although reactive strategies require comparably little information, examples such as Tit-for-Tat [5] and Generous Tit-for-Tat [14, 15] indicate that they are remarkably successful in sustaining cooperation. When players take into account both their own and their co-player's previous move, we speak of memory-1 strategies. Among memory-1 strategies, players often learn to adopt a simple rule termed win-stay lose-shift [16, 17]. Similarly successful strategies can be identified among memory-2 strategies [18], and more generally among memory-$n$ strategies [19, 20].

To explore the impact of memory on the evolution of cooperation, much of the existing literature assumes that all population members use the same strategy space and thus have the same memory capacity. This work suggests that cooperation is the more likely to evolve the more rounds players remember [18–22]. When interactions take place in an entire population, longer memory can also help players to learn and to classify their co-players' strategies. This can be advantageous if it allows players to adapt their own strategy to the population's overall strategy distribution [23–25].

Interestingly however, there is in contrast less work on how likely cooperation is to evolve when co-players differ in how much they remember, and whether larger strategy spaces (using more memory) can emerge in the first place. Press and Dyson [26] argued that no player can gain an advantage in a repeated prisoner's dilemma by switching to a higher memory-strategy when interacting with an opponent using a memory-1 strategy in a fixed game. This finding led to a stronger focus on memory-1 strategies [27–34]. On the other hand, studies on the co-evolution of behavior and memory on the same timescale in the context of multiplayer games suggest that when games are played by few individuals, longer memories can evolve more easily [35]. This increased memory capacity in turn facilitates cooperation, as players gain access to a greater set of evolutionary robust cooperative strategies.

Additionally, there is experimental literature on the topic of memory capacity and cooperation supporting the conclusion that a higher capacity for remembering the game's history makes players more successful. Such work studied the correlation between cognitive load, memory or cognitive ability, and performance in a repeated prisoner's dilemma [36, 37]. It suggests in fact that a higher cognitive load, for example by giving players an additional memorization task, constrains the human working memory. This then affects decision making,

leading to players adopting simpler strategies or behaving less strategically in the game. Consequently, players with a high external cognitive load are less successful in a repeated game than those under low load, due to not being able to commit as many cognitive resources to their play.

Instead of assuming that all players use the same memory capacity [18–21], or that memory capacities evolve at a similar pace as the players' strategies [35], here we study an alternative framework. In our approach, memory capacities and strategies evolve at different time scales. In the short run, each player's strategy space is fixed. Players with possibly different strategy spaces interact in repeated donation games [2]. They update their strategies given their individual constraints. The respective results can be interpreted as a tournament among players with access to different strategy spaces.

In a second step, we consider the long run dynamics. Here, the players' strategy spaces can change in time. To this end, we use the payoffs of the previous tournaments as a proxy for the payoff of a given strategy space. Using these payoffs, we can study "supergames" between different strategy spaces. By exploring the corresponding replicator dynamics [38], we examine which strategy space is favored by evolution. A similar argument of separation of timescales has been used previously in the study of altruism in animal behavior [39] and the evolution of social dominance [40].

Intuition and previous work [23–25] suggest that larger memory capacities should be advantageous: since memory is the main resource in repeated interactions, a player with longer memory should retain a payoff advantage over a shorter-memory opponent. However, in our model this is not necessarily the case. Instead we find that when cooperation is comparably costly or when selection is weak, smaller strategy spaces (lower memory) tend to succeed. In this parameter range, cooperation is generally difficult to achieve. As a consequence, small strategy spaces are favored because they allow players to more quickly discover strategies that defect. In such a case, the resulting competition among players with different strategy spaces can constitute a higher-order social dilemma: each player individually benefits from smaller strategy space, even if this ultimately favors outcomes that are detrimental to both.

To show these results, we focus in the following on three different strategy spaces that have different requirements for memory capacity: unconditional strategies, reactive strategies, and memory-1 strategies.

## Results

### Repeated games among players with different memory capacity

We consider two players who engage in an infinitely iterated donation game, a special case of a Prisoner's Dilemma [2]. In each round, players can choose among two possible actions, cooperation ($C$) and defection ($D$). A cooperating player pays a cost $c$ to confer a benefit $b$ to the co-player. A defecting co-player pays no cost and confers no benefit. This results in the payoff matrix

$$
\begin{array}{cc}
 & \begin{array}{cc} C & D \end{array} \\
\begin{array}{c} C \\ D \end{array} & \begin{pmatrix} b-c & -c \\ b & 0 \end{pmatrix}
\end{array}.
\tag{1}
$$

Under the usual assumption that $b > c > 0$, both players prefer mutual cooperation to mutual defection, yet each player has a temptation to defect. Players independently choose their action in each round, governed by their respective strategies. These strategies prescribe actions based on the history of the game: individuals can recall previous outcomes of their

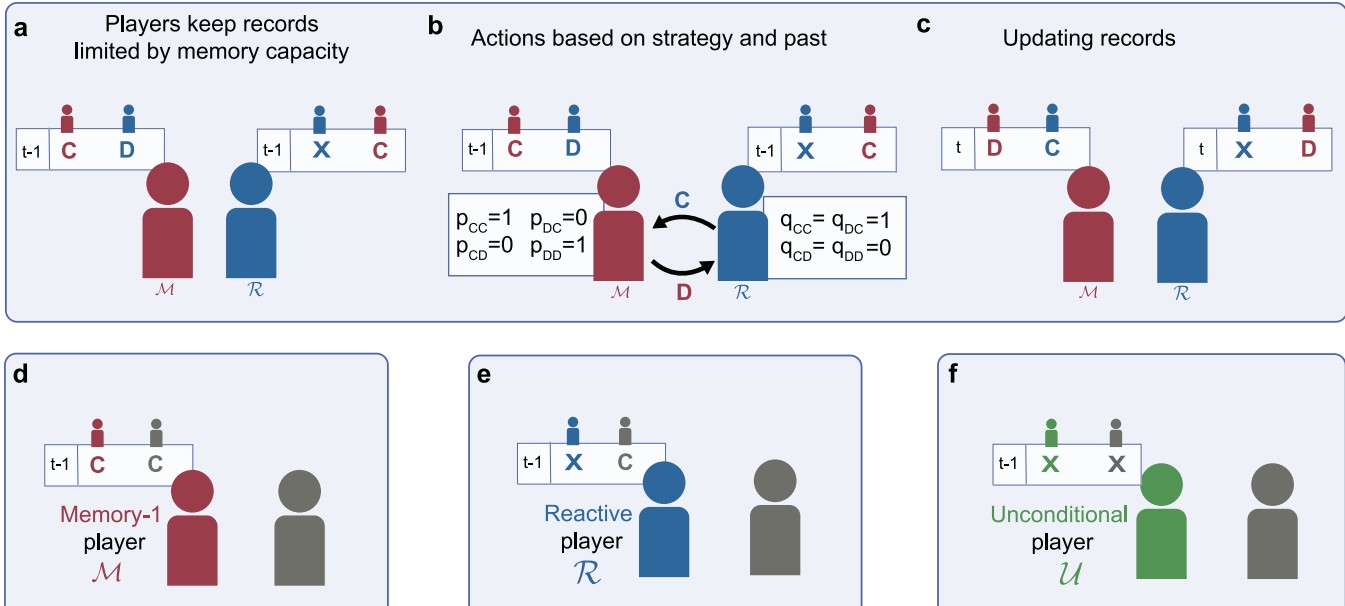

**Fig 1. Direct reciprocity among players with different memory capacities. a-c**, Models of direct reciprocity assume that two or more players repeatedly engage in a social dilemma. In each round $t$, players can either cooperate ($C$) or defect ($D$). To make these decisions, players keep a record of what happened in previous rounds. Based on these records, they decide whether or not to cooperate in the current round. After each round, players update their private records. These records may be constrained by how much players remember. Here we distinguish three memory spaces. **d**, Memory-1 players ($\mathcal{M}$) remember both their own and their co-player's previous action. **e**, Reactive players ($\mathcal{R}$) only remember their co-player's previous action. **f**, Unconditional players ($\mathcal{U}$) keep no records at all. The example in the first row illustrates an interaction between a memory-1 player with strategy *WSLS* [16] against a reactive player with strategy *TFT* [5].

encounters with an opponent (Fig 1a); these outcomes affect the actions they choose next (Fig 1b); after each such interaction, individuals update their records for the next round (Fig 1c).

We assume that players have limited memory capacity. This determines how many past events, i.e. which history length, players can take into account when making their decision. In the following, we suppose that players remember at most the outcome of the previous round. We refer to the corresponding set of strategies as the space of memory-1 strategies, $\mathcal{M}$ (Fig 1d):

$$\mathcal{M} = \{ \; \mathbf{p} = (p_{CC}, p_{CD}, p_{DC}, p_{DD}) \mid 0 \leq p_{ij} \leq 1 \; \}. \tag{2}$$

These strategies take into account both players' actions in the previous round for decision making. An entry $p_{ij}$ refers to the player's probability to cooperate in the next round, if in the previous round the focal player chose action $i$ and the co-player chose action $j$. One example of such a strategy is win-stay lose-shift, $WSLS = (1, 0, 0, 1)$, which cooperates if and only if either both players cooperated in the previous round or if no one did [16]. Past work [16, 20] indicates that $WSLS$ frequently evolves among memory-1 players, and that it is robust against invasion by any other strategy when $c < b/2$.

In addition to memory-1 strategies, there is also a substantial literature on players who only remember the co-player's previous action [14, 41–45]. We refer to the respective set of reactive strategies as $\mathcal{R}$ (Fig 1e):

$$\mathcal{R} = \{ \; \mathbf{p} \in \mathcal{M} \mid p_{CC} = p_{DC}, \; p_{DC} = p_{DD} \; \}. \tag{3}$$

Strategies in $\mathcal{R}$ can be represented as those memory-1 strategies that do not condition their behavior on their own previous move. An important example of a reactive strategy is Tit-for-Tat [5] (*TFT*), for which $p_{XC} = 1$ and $p_{XD} = 0$. Another well-known example is Generous Tit-for-Tat [14, 15] (*GTFT*), for which $p_{XC} = 1$ and $p_{XD} = 1 - c/b$.

Finally, we also consider players who do not remember any previous events at all. Such players choose among all unconditional strategies $\mathcal{U}$ (Fig 1f), which can be represented as those memory-1 strategies that play the same action irrespective of the previous outcome. Formally, we have

$$\mathcal{U} = \{ \; \mathbf{p} \in \mathcal{M} \mid p_{CC} = p_{DC} = p_{DC} = p_{DD} \; \}. \tag{4}$$

The set $\mathcal{U}$ contains, for example, *ALLC* (for which $p_{XX} = 1$) and *ALLD* (for which $p_{XX} = 0$). By definition, $\mathcal{U} \subset \mathcal{R} \subset \mathcal{M}$; any strategy that can be implemented by a low-memory player can also be implemented by a higher-memory player.

We note that the general framework can be easily extended to larger memory capacities. However, the size of the respective strategy spaces increases exponentially [20]. Additionally, apart from yielding good approximations to actual human and animal behavior [46, 47], the space of memory-1 strategies (and its subspaces) has the advantage that payoffs can be computed explicitly. More specifically, suppose the first player adopts the strategy $\mathbf{p}$, and the second player uses the strategy $\mathbf{q}$, with $\mathbf{p}, \mathbf{q} \in \{\mathcal{M}, \mathcal{R}, \mathcal{U}\}$. Then the players' payoffs in the repeated donation game can be calculated by representing the game as a Markov chain. We describe this approach in the **Methods**.

## Learning among players with fixed memory capacities

To explore how the players' memory capacity affects the evolutionary dynamics, we consider two given players, player 1 and player 2. Each player $i$ is constrained to choose strategies from a fixed strategy space $\mathcal{S}_i \in \{\mathcal{M}, \mathcal{R}, \mathcal{U}\}$. In particular, the two players may differ in their memory capacities; in that case $\mathcal{S}_1 \neq \mathcal{S}_2$. Over time, both players explore their strategies to adapt to their opponent. To model how players adapt their strategies, we use introspection dyamics. This process has been previously used to model evolution in asymmetric repeated games [48, 49].

According to introspection dynamics, the two players initially start out with random strategies $\mathbf{p}_1 \in \mathcal{S}_1$ and $\mathbf{p}_2 \in \mathcal{S}_2$. In each time step of the learning process, one of the two players, say player $i$, is randomly picked and given the opportunity to revise its strategy. To this end, the player experiments with a randomly chosen new strategy $\mathbf{p}_i' \in \mathcal{S}_i$. If the player's original strategy yields a payoff of $\pi_i$ and the new strategy yields the payoff $\pi_i'$, player $i$ switches to the new strategy with probability

$$\varrho = \frac{1}{1 + e^{-\beta(\pi_i' - \pi_i)}}, \tag{5}$$

Here $\beta \geq 0$ is the strength of selection. It measures to which extent strategies are chosen based on the payoff they yield. If $\beta$ is small, we speak of weak selection; in this case, payoffs only have little influence on the strategies players adopt. Instead, strategy selection is mostly driven by random chance events (noise). If $\beta$ is large, we speak of strong selection. In that case, players only switch to an alternative strategy if it improves their payoff.

We repeat this elementary strategy updating step for many time steps $t$, with $t \in \{1, \ldots, T\}$. As a result, we obtain a sequence of payoffs $(\pi_{ij}(t))_{t=1}^{T}$. Here, $\pi_{ij}(t)$ is player $i$'s payoff against player $j$ after $t$ time steps, and $T$ is the total number of time steps during which learning takes

place. Based on this sequence, we compute the time average

$$\bar{\pi}_{\mathcal{S}_i \mathcal{S}_j} = \frac{1}{T} \sum_{t=1}^{T} \pi_{ij}(t). \tag{6}$$

We interpret $\bar{\pi}_{ij}$ as the average payoff of a player with memory capacity $\mathcal{S}_i$ against a co-player with memory capacity $\mathcal{S}_j$. In the following, we explore how these payoffs depend on the players' strategy spaces, on the costs and benefits of cooperation, and on the selection strength. To this end, we set without loss of generality $b := 1$, such that the value of $c$ reflects the cost-to-benefit ratio $c/b$ of the game.

## Tournament results

We first use tournaments to explore pairwise competitions between players with different memory capacities. For each possible combination $\mathcal{S}_i$ and $\mathcal{S}_j$, we use simulations to explore the introspection dynamics when two players with the respective memory spaces compete. We compare the performance of the two memory spaces by comparing the resulting payoffs $\bar{\pi}_{\mathcal{S}_i \mathcal{S}_j}$ and $\bar{\pi}_{\mathcal{S}_j \mathcal{S}_i}$, as defined by Eq (6).

To gain some intuition, we start with a scenario in which cooperation is relatively cheap ($c/b = 0.2$), and in which selection is comparably strong ($\beta = 100$, see also Table 1). When we pitch a memory-1 player against an unconditional player, we observe that the higher-memory player outcompetes its lower-memory opponent, with $\bar{\pi}_{\mathcal{M},\mathcal{U}} \approx 0.24$ compared to $\bar{\pi}_{\mathcal{U},\mathcal{M}} \approx 0.18$. The same qualitative result holds when it is a reactive player that faces an unconditional opponent, with $\bar{\pi}_{\mathcal{R},\mathcal{U}} \approx 0.25$ compared to $\bar{\pi}_{\mathcal{U},\mathcal{R}} \approx 0.14$. Surprisingly, however, the ranking reverses when a memory-1 player faces a reactive opponent, with $\bar{\pi}_{\mathcal{M},\mathcal{R}} \approx 0.38$ compared to $\bar{\pi}_{\mathcal{R},\mathcal{M}} \approx 0.40$. Here it is the reactive player with lower memory capacity who wins the competition. Overall, the reactive player thus wins two out of two pairwise competitions, the memory-1 player wins one competition, and the unconditional player wins no competition.

In a next step, we have explored the robustness of these findings by systematically varying the cost-to-benefit ratio between 0.1 and 0.9 (keeping the strength of selection fixed at $\beta = 100$). We observe that the overall results remain unchanged (Tables 1 and 2). In particular, for

**Table 1. Pairwise competitions of players with different memory spaces.** We consider the learning dynamics among players with different memory spaces. Players either use memory-1 strategies ($\mathcal{M}$), reactive strategies ($\mathcal{R}$) or unconditional strategies ($\mathcal{U}$). Within their respective memory space, players adapt their strategies to their opponent using introspection dynamics [48, 49] with a selection strength of $\beta = 100$. For each combination of memory spaces, we compute the players' average payoffs according to Eq (6). The winners of these pairwise comparisons are shown in bold. We find that reactive players outperform both memory-1 opponents and unconditional opponents for all considered cost values. Simulations are run for $T = 10^9$ time steps.

| Cost $c$ | $\mathcal{M} : \mathcal{R}$ | | $\mathcal{M} : \mathcal{U}$ | | $\mathcal{R} : \mathcal{U}$ | | $(b - c)$ |
|---|---|---|---|---|---|---|---|
| | $\bar{\pi}_{\mathcal{M},\mathcal{R}}$ | $\bar{\pi}_{\mathcal{R},\mathcal{M}}$ | $\bar{\pi}_{\mathcal{M},\mathcal{U}}$ | $\bar{\pi}_{\mathcal{U},\mathcal{M}}$ | $\bar{\pi}_{\mathcal{R},\mathcal{U}}$ | $\bar{\pi}_{\mathcal{U},\mathcal{R}}$ | |
| 0.1 | 0.559 | **0.566** | **0.370** | 0.289 | **0.375** | 0.235 | 0.9 |
| 0.2 | 0.377 | **0.402** | **0.240** | 0.177 | **0.246** | 0.138 | 0.8 |
| 0.3 | 0.248 | **0.288** | **0.147** | 0.116 | **0.156** | 0.086 | 0.7 |
| 0.4 | 0.156 | **0.205** | **0.087** | 0.077 | **0.098** | 0.054 | 0.6 |
| 0.5 | 0.091 | **0.142** | 0.049 | **0.052** | **0.060** | 0.034 | 0.5 |
| 0.6 | 0.048 | **0.094** | 0.026 | **0.034** | **0.036** | 0.020 | 0.4 |
| 0.7 | 0.020 | **0.058** | 0.012 | **0.022** | **0.020** | 0.011 | 0.3 |
| 0.8 | 0.004 | **0.033** | 0.0035 | **0.014** | **0.01** | 0.006 | 0.2 |
| 0.9 | -0.004 | **0.017** | -0.001 | **0.008** | **0.004** | 0.002 | 0.1 |

**Table 2. Wins and scores in pairwise tournaments.** To interpret the results of Table 1, we (*i*) count how often a memory space wins a pairwise competition, and (*ii*) we compute the memory space's score by adding up its payoff against the two other memory spaces. With respect to both measures, we find again that reactive players outperform players using the other two memory spaces. In general, however see that the rankings by wins and total score can differ. For example, for $c = 0.5$ and $c = 0.6$, $\mathcal{U}$ wins more often than $\mathcal{M}$, but ranks last in terms of total score. Parameters are the same as in Table 1.

| Cost $c$ | Wins | | | | Score | | | | $(b - c)$ |
|---|---|---|---|---|---|---|---|---|---|
| | $\mathcal{M}$ | $\mathcal{R}$ | $\mathcal{U}$ | Ranking | $\mathcal{M}$ | $\mathcal{R}$ | $\mathcal{U}$ | Ranking | |
| 0.1 | 1 | **2** | 0 | $\mathcal{R}, \mathcal{M}, \mathcal{U}$ | 0.929 | **0.941** | 0.85 | $\mathcal{R}, \mathcal{M}, \mathcal{U}$ | 0.9 |
| 0.2 | 1 | **2** | 0 | $\mathcal{R}, \mathcal{M}, \mathcal{U}$ | 0.617 | **0.648** | 0.315 | $\mathcal{R}, \mathcal{M}, \mathcal{U}$ | 0.8 |
| 0.3 | 1 | **2** | 0 | $\mathcal{R}, \mathcal{M}, \mathcal{U}$ | 0.395 | **0.444** | 0.202 | $\mathcal{R}, \mathcal{M}, \mathcal{U}$ | 0.7 |
| 0.4 | 1 | **2** | 0 | $\mathcal{R}, \mathcal{M}, \mathcal{U}$ | 0.243 | **0.303** | 0.131 | $\mathcal{R}, \mathcal{M}, \mathcal{U}$ | 0.6 |
| 0.5 | 0 | **2** | 1 | $\mathcal{R}, \mathcal{U}, \mathcal{M}$ | 0.140 | **0.202** | 0.086 | $\mathcal{R}, \mathcal{M}, \mathcal{U}$ | 0.5 |
| 0.6 | 0 | **2** | 1 | $\mathcal{R}, \mathcal{U}, \mathcal{M}$ | 0.074 | **0.13** | 0.054 | $\mathcal{R}, \mathcal{M}, \mathcal{U}$ | 0.4 |
| 0.7 | 0 | **2** | 1 | $\mathcal{R}, \mathcal{U}, \mathcal{M}$ | 0.032 | **0.078** | 0.033 | $\mathcal{R}, \mathcal{U}, \mathcal{M}$ | 0.3 |
| 0.8 | 0 | **2** | 1 | $\mathcal{R}, \mathcal{U}, \mathcal{M}$ | 0.008 | **0.043** | 0.020 | $\mathcal{R}, \mathcal{U}, \mathcal{M}$ | 0.2 |
| 0.9 | 0 | **2** | 1 | $\mathcal{R}, \mathcal{U}, \mathcal{M}$ | -0.005 | **0.021** | 0.010 | $\mathcal{R}, \mathcal{U}, \mathcal{M}$ | 0.1 |

all $c/b$ ratios considered, the reactive player outcompetes both other strategy spaces. For the pairwise competition between a memory-1 player and an unconditional opponent, the memory-1 players win for $c/b < 0.5$; for $c \geq 0.5$, the memory-1 player ranks last (For no parameter combination, we observe a non-transitive ranking, where each memory space wins exactly one pairwise competition in a rock paper scissors fashion).

While the previous tournaments take into account how well different strategy spaces perform against each other, they ignore a memory space's performance against itself. To compute these *self payoffs*, we run additional simulations in which two players with the same memory space interact (Table 3). As one may expect from the previous literature [18–21], we find that with respect to self payoffs, a larger memory capacity tends to be beneficial: two memory-1 players get a larger payoff than two reactive players, who in turn get more than two unconditional players.

To get an overall measure for a strategy space's success, we also compute a *combined score*. For this combined score, we add up a strategy space's average payoff against all three possible strategy spaces (including itself). Based on this combined score, we find that memory-1 strategies outperform the other two memory spaces when cooperation is cheap (when $c \leq 0.3$). Otherwise, for intermediate to high costs, reactive strategies again come out first.

We have run similar tournaments for various selection strengths, $\beta \in \{1, 10, 100, 1000\}$; the respective results are displayed in **Tables A-I in** S1 Data. In Table 4, we provide an overview. Similar to before, we consider four complementary measures for a memory space's success: (*i*) the number of wins in a pairwise competition against another memory space, (*ii*) the overall payoff (score) against the other two memory spaces, (*iii*) the memory space's self-payoff, and (*iv*) the combined score, as defined above. When selection is weak ($\beta = 1$), we find that memory-1 strategies typically yield the highest self payoff, yet unconditional strategies succeed with respect to all other measures. On the other extreme, for strong selection ($\beta = 1000$), we find that memory-1 strategies still yield the highest self-payoff. However, now they can also perform well with respect to the other measures, provided cooperation is sufficiently cheap. Only when cooperation costs exceed a certain threshold, reactive strategies succeed. This threshold for reactive strategies to succeed is lowest for the number of wins (when $c \geq 0.2$), intermediate for the score (when $c \geq 0.3$), and highest for the combined score (when $c \geq 0.8$), see Table 4.

We conclude that although memory-1 players tend to obtain the best self payoff, they are typically outcompeted by co-players with lower memory, especially when selection is weak or

**Table 3. Self scores and combined scores in pairwise tournaments.** In addition to the wins and the scores considered in Table 2, we consider two additional measures for a memory space's success. A memory space's self payoff is the payoff two players with the respective memory space obtain against each other. The combined score is the sum of the memory space's score and its self payoff. Again, winners are marked in bold face. Across all cooperation costs, we find that memory-1 strategies yield the largest self payoff. They also achieve the largest combined score if $c \leq 0.3$; otherwise, for $c \geq 0.4$, reactive strategies succeed. Parameters are the same as in Table 1.

| Cost $c$ | Self payoff | | | Combined score | | | | $(b - c)$ |
|---|---|---|---|---|---|---|---|---|
| | $\bar{\pi}_{\mathcal{M},\mathcal{M}}$ | $\bar{\pi}_{\mathcal{R},\mathcal{R}}$ | $\bar{\pi}_{\mathcal{U},\mathcal{U}}$ | $\mathcal{M}$ | $\mathcal{R}$ | $\mathcal{U}$ | Ranking | |
| 0.1 | **0.664** | 0.538 | 0.090 | **1.593** | 1.478 | 0.613 | $\mathcal{M}, \mathcal{R}, \mathcal{U}$ | 0.9 |
| 0.2 | **0.478** | 0.382 | 0.040 | **1.095** | 1.030 | 0.354 | $\mathcal{M}, \mathcal{R}, \mathcal{U}$ | 0.8 |
| 0.3 | **0.322** | 0.267 | 0.023 | **0.717** | 0.711 | 0.226 | $\mathcal{M}, \mathcal{R}, \mathcal{U}$ | 0.7 |
| 0.4 | **0.210** | 0.180 | 0.015 | 0.453 | **0.483** | 0.146 | $\mathcal{R}, \mathcal{M}, \mathcal{U}$ | 0.6 |
| 0.5 | **0.135** | 0.114 | 0.010 | 0.276 | **0.316** | 0.095 | $\mathcal{R}, \mathcal{M}, \mathcal{U}$ | 0.5 |
| 0.6 | **0.084** | 0.067 | 0.0067 | 0.1582 | **0.196** | 0.061 | $\mathcal{R}, \mathcal{M}, \mathcal{U}$ | 0.4 |
| 0.7 | **0.049** | 0.035 | 0.0043 | 0.0806 | **0.113** | 0.038 | $\mathcal{R}, \mathcal{M}, \mathcal{U}$ | 0.3 |
| 0.8 | **0.024** | 0.016 | 0.0025 | 0.0321 | **0.059** | 0.022 | $\mathcal{R}, \mathcal{M}, \mathcal{U}$ | 0.2 |
| 0.9 | **0.009** | 0.005 | 0.001 | 0.004 | **0.026** | 0.011 | $\mathcal{R}, \mathcal{U}, \mathcal{M}$ | 0.1 |

when cooperation is costly. To make sense of these observations, we note that both weak selection and costly cooperation make full cooperation difficult to sustain. In such a scenario, it can be important for players to quickly switch from a fully cooperative strategy to a strategy that effectively defects. However, for memory-1 players it is comparably difficult to come up with extreme behaviors. Instead, most random memory-1 strategies lead to intermediate

**Table 4. Tournament winners for different rankings and selection strengths.** We summarize our static results by showing the tournament's winner for various costs and selection strengths, with respect to the four different rankings (i) number of wins, (ii) score, (iii) self payoff, and (iv) combined score. Memory-1 strategies typically obtain the highest self-payoff. However, they only succeed in the other rankings when cooperation is cheap (small $c$) and when selection is sufficiently strong (large $\beta$).

| $c$ | $\beta = 1$ | | | | $\beta = 10$ | | | |
|---|---|---|---|---|---|---|---|---|
| | Wins | Score | Self | Comb. | Wins | Score | Self | Comb. |
| 0.1 | $\mathcal{U}$ | $\mathcal{U}$ | $\mathcal{R}$ | $\mathcal{U}$ | $\mathcal{U}$ | $\mathcal{U}$ | $\mathcal{R}$ | $\mathcal{R}$ |
| 0.2 | $\mathcal{U}$ | $\mathcal{U}$ | $\mathcal{M}$ | $\mathcal{U}$ | $\mathcal{U}$ | $\mathcal{U}$ | $\mathcal{R}$ | $\mathcal{R}$ |
| 0.3 | $\mathcal{U}$ | $\mathcal{U}$ | $\mathcal{M}$ | $\mathcal{U}$ | $\mathcal{U}$ | $\mathcal{U}$ | $\mathcal{M}$ | $\mathcal{R}$ |
| 0.4 | $\mathcal{U}$ | $\mathcal{U}$ | $\mathcal{M}$ | $\mathcal{U}$ | $\mathcal{U}$ | $\mathcal{U}$ | $\mathcal{M}$ | $\mathcal{R}$ |
| 0.5 | $\mathcal{U}$ | $\mathcal{U}$ | $\mathcal{M}$ | $\mathcal{U}$ | $\mathcal{U}$ | $\mathcal{U}$ | $\mathcal{M}$ | $\mathcal{R}$ |
| 0.6 | $\mathcal{U}$ | $\mathcal{U}$ | $\mathcal{M}$ | $\mathcal{U}$ | $\mathcal{U}$ | $\mathcal{U}$ | $\mathcal{M}$ | $\mathcal{R}$ |
| 0.7 | $\mathcal{U}$ | $\mathcal{U}$ | $\mathcal{M}$ | $\mathcal{U}$ | $\mathcal{U}$ | $\mathcal{U}$ | $\mathcal{M}$ | $\mathcal{R}$ |
| 0.8 | $\mathcal{U}$ | $\mathcal{U}$ | $\mathcal{M}$ | $\mathcal{U}$ | $\mathcal{U}$ | $\mathcal{U}$ | $\mathcal{M}$ | $\mathcal{U}$ |
| 0.9 | $\mathcal{U}$ | $\mathcal{U}$ | $\mathcal{M}$ | $\mathcal{U}$ | $\mathcal{U}$ | $\mathcal{U}$ | $\mathcal{M}$ | $\mathcal{U}$ |

| $c$ | $\beta = 100$ | | | | $\beta = 1000$ | | | |
|---|---|---|---|---|---|---|---|---|
| | Wins | Score | Self | Comb. | Wins | Score | Self | Comb. |
| 0.1 | $\mathcal{R}$ | $\mathcal{R}$ | $\mathcal{M}$ | $\mathcal{M}$ | $\mathcal{M}$ | $\mathcal{M}$ | $\mathcal{M}$ | $\mathcal{M}$ |
| 0.2 | $\mathcal{R}$ | $\mathcal{R}$ | $\mathcal{M}$ | $\mathcal{M}$ | $\mathcal{R}$ | $\mathcal{M}$ | $\mathcal{M}$ | $\mathcal{M}$ |
| 0.3 | $\mathcal{R}$ | $\mathcal{R}$ | $\mathcal{M}$ | $\mathcal{M}$ | $\mathcal{R}$ | $\mathcal{R}$ | $\mathcal{M}$ | $\mathcal{M}$ |
| 0.4 | $\mathcal{R}$ | $\mathcal{R}$ | $\mathcal{M}$ | $\mathcal{R}$ | $\mathcal{R}$ | $\mathcal{R}$ | $\mathcal{M}$ | $\mathcal{M}$ |
| 0.5 | $\mathcal{R}$ | $\mathcal{R}$ | $\mathcal{M}$ | $\mathcal{R}$ | $\mathcal{R}$ | $\mathcal{R}$ | $\mathcal{M}$ | $\mathcal{M}$ |
| 0.6 | $\mathcal{R}$ | $\mathcal{R}$ | $\mathcal{M}$ | $\mathcal{R}$ | $\mathcal{R}$ | $\mathcal{R}$ | $\mathcal{M}$ | $\mathcal{M}$ |
| 0.7 | $\mathcal{R}$ | $\mathcal{R}$ | $\mathcal{M}$ | $\mathcal{R}$ | $\mathcal{R}$ | $\mathcal{R}$ | $\mathcal{M}$ | $\mathcal{M}$ |
| 0.8 | $\mathcal{R}$ | $\mathcal{R}$ | $\mathcal{M}$ | $\mathcal{R}$ | $\mathcal{R}$ | $\mathcal{R}$ | $\mathcal{M}$ | $\mathcal{R}$ |
| 0.9 | $\mathcal{R}$ | $\mathcal{R}$ | $\mathcal{M}$ | $\mathcal{R}$ | $\mathcal{R}$ | $\mathcal{R}$ | $\mathcal{M}$ | $\mathcal{R}$ |

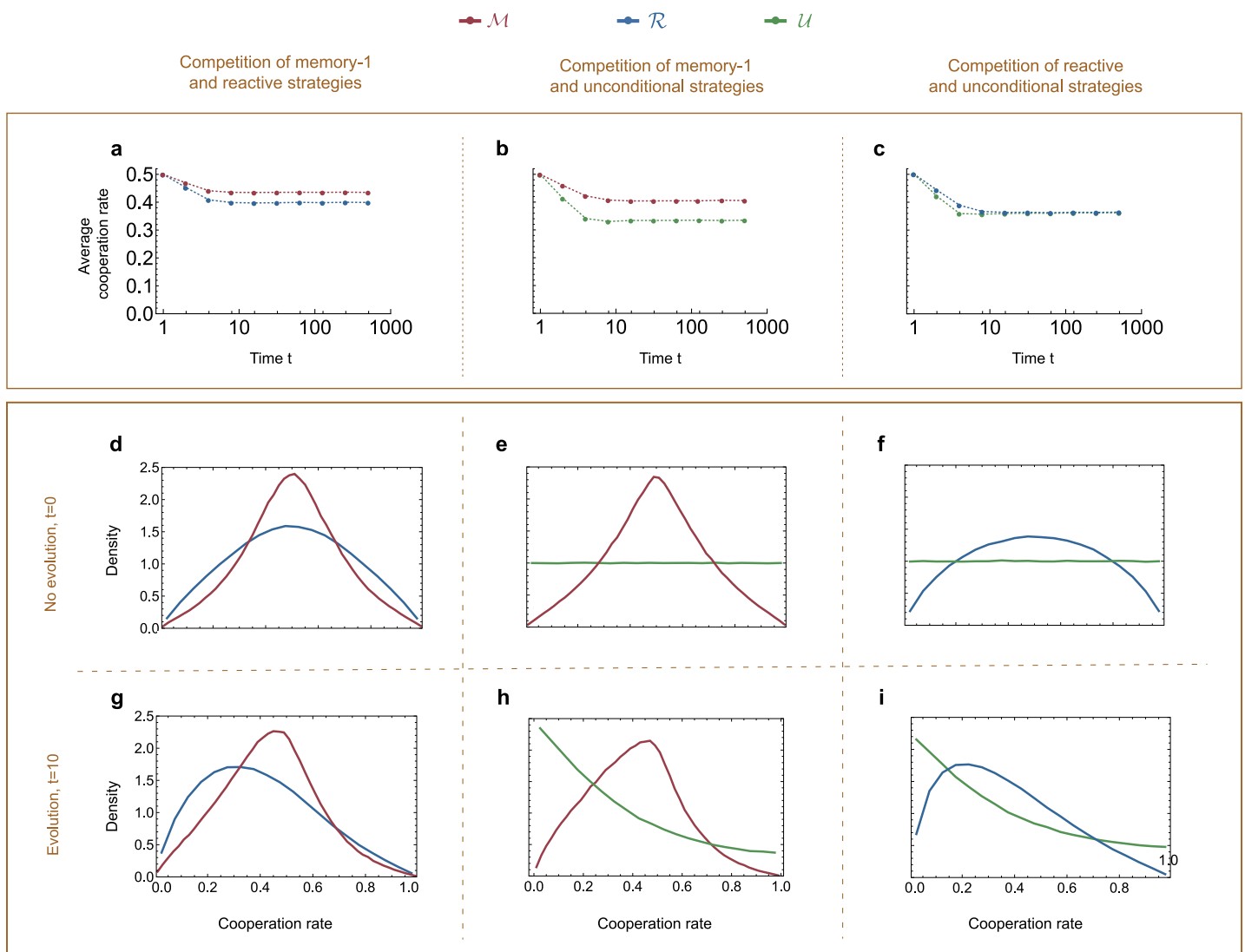

**Fig 2. Lower-memory strategies are more likely to discover strategies with extreme cooperation rates.** Here, we study the distribution of the players' cooperation rates when two players with different memory capacity interact ($\mathcal{M}$ vs $\mathcal{R}$, $\mathcal{M}$ vs $\mathcal{U}$, $\mathcal{R}$ vs $\mathcal{U}$). In panels **a–c**, we show that the mean of this distribution stabilizes after at most $t = 10$ timesteps, for each combination of memory spaces. In **d-f**, we present this distribution in the very beginning of the process, when players choose their strategies uniformly at random from their respective memory space. In **g-i**, we show how the distribution of cooperation rates changes after 10 time steps of introspection dynamics (for $c = 0.5$ and $\beta = 10$), which is the time it takes for the difference in average cooperation rates to stabilize. In both cases, we observe that players with lower memory capacities are more likely to choose strategies with extremal cooperation rates (a cooperation rate close to zero or one). To create this graph, we have randomly sampled $10^6$ pairs of strategies from the respective memory spaces. For each pair, we have then simulated $t = 10$ time steps of introspection dynamics. The curves show the result when we bin the players' cooperation rates in steps of 0.02 and renormalize (such that the area under each curve is one). The above plots show marginal distributions for each memory-space. In contrast, S5 and S6 Figs show joint distributions for each possible combination of memory spaces.

cooperation rates (Fig 2). In comparison, both reactive and unconditional strategies are better able to discover extreme strategies with either very low or very high cooperation rates. According to our numerical results, this enhanced flexibility can offer an advantage to lower memory spaces, especially in parameter regimes in which stable cooperation is rare.

## Evolutionary dynamics of memory spaces

After this static tournament approach to study games among players with different memory capacities, we allow the players' memory capacities themselves to evolve in time. To this end,

we imagine a large and well mixed population of players. The dynamics takes place on two time scales.

In the short run, the players' memory capacity is fixed. Let $x_{\mathcal{M}}$, $x_{\mathcal{R}}$, and $x_{\mathcal{U}}$ denote the respective fractions of players with given memory capacity. These players are then randomly matched to engage in pairwise tournaments, as in the previous section. As before, if a player with memory space $\mathcal{S}_i$ interacts with a co-player with memory space $\mathcal{S}_j$, the resulting payoff is $\bar{\pi}_{\mathcal{S}_i \mathcal{S}_j}$, as defined by Eq (6). These payoffs can be assembled in a $3 \times 3$ payoff matrix. Here, each row corresponds to the memory space $\mathcal{S}_i$ of the focal player, and each column refers to the memory space $\mathcal{S}_j$ of the opponent, with $\mathcal{S}_i, \mathcal{S}_j \in \{\mathcal{M}, \mathcal{R}, \mathcal{U}\}$. The entries of the matrix reflect the focal player's payoff against the given opponent.

In the long run, the fractions of players with a given memory capacity may change, depending on how successful respective players turn out to be. While this change in the players' memory capacities could be captured with several dynamical models, here we use standard replicator dynamics [50] (for details, see Methods). We note however that many of our results are independent of the specific dynamics we consider. For example, we sometimes find that one memory space dominates another. In that case, replicator dynamics leads to the extinction of the dominated strategy space, but so does any other "payoff-monotone" dynamics [51].

In replicator dynamics, if a player's memory space yields a payoff above the average payoff in the population, this player is more likely to reproduce. In the special case that players can only choose among two possible memory spaces, there are three generic scenarios for the dynamics (Fig 3a): (*i*) One memory space is globally stable (dominance); (*ii*) both memory spaces are locally stable (bistability); and (*iii*) neither memory space is globally stable, but instead populations converge to a stable mixture of players with different memory capacities (coexistence). In the general case that players can choose among three memory spaces the dynamics can be more complex, but it is still possible to classify all possible qualitative behaviors [52].

To exemplify our approach, we first consider a scenario in which cooperation is rather costly ($c = 0.6$) and in which selection is comparably weak ($\beta = 10$, Fig 3b). For the given parameter values, the pairwise payoffs can be recovered from the previous tournament results (**Tables D, F in** S1 Data); the respective payoff matrix is given by

$$
\begin{array}{c}
\\
\mathcal{M} \\
\mathcal{R} \\
\mathcal{U}
\end{array}
\begin{array}{c}
\begin{array}{ccc}
\mathcal{M} & \mathcal{R} & \mathcal{U}
\end{array} \\
\begin{pmatrix}
0.154 & 0.106 & 0.047 \\
0.184 & 0.140 & 0.099 \\
0.187 & 0.121 & 0.066
\end{pmatrix},
\end{array}
\tag{7}
$$

Based on this payoff matrix, we can analyze the resulting replicator dynamics. To this end, we first consider the boundaries of the state space (Fig 3c–3e). Here, only two of the three memory spaces compete; the third memory space is absent from the population. If the population only consists of reactive and unconditional players, we find that the reactive strategies are globally stable (Fig 3c). On the other hand, if memory-1 strategies compete with either unconditional strategies (Fig 3d) or reactive strategies (Fig 3e), it is in each case the lower memory space that is globally stable. In a next step, we explore the dynamics when all three memory spaces compete simultaneously. In this case, we find that reactive strategies succeed: for any given initial mixture that consists of all three strategy spaces, the dynamics eventually converges to a monomorphic population of reactive players (Fig 3f).

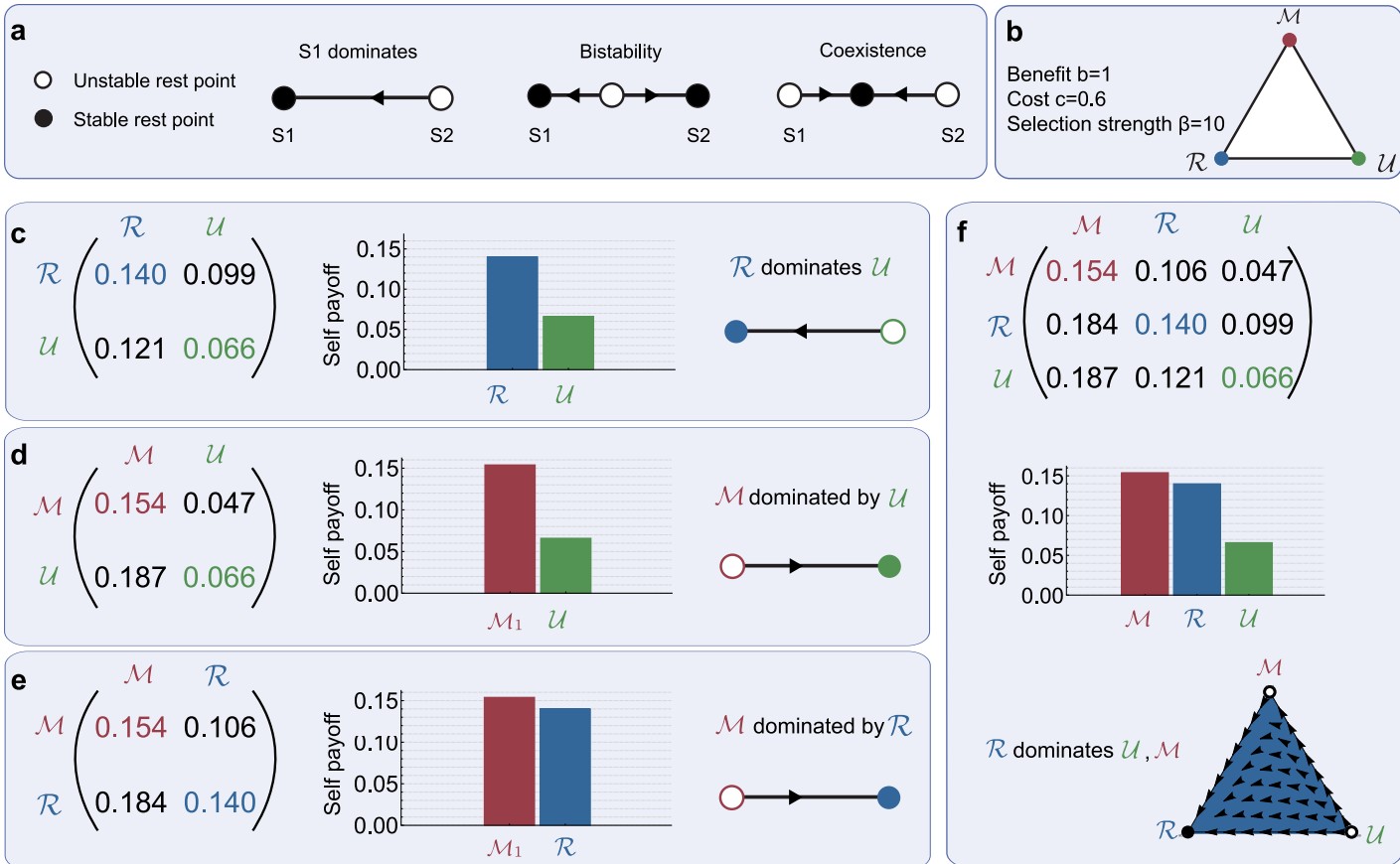

**Fig 3. Evolutionary dynamics of memory spaces.** To explore the evolution of different memory capacities, we use replicator dynamics [50]. Members of a population can have one of three different memory spaces, memory-1 ($\mathcal{M}$), reactive ($\mathcal{R}$), or unconditional ($\mathcal{U}$). The fraction of population members with a given memory space changes in time, depending on whether players with this memory space obtain an expected payoff above average. **a,** When only two memory spaces compete, there are three possible dynamics: either one space is globally stable (dominance), each space is locally stable (bistability), or the two spaces form a stable mixture (coexistence). **b,** Here we illustrate our approach by considering an environment in which cooperation is comparably costly and where selection is relatively weak. **c-e,** We first analyze the replicator dynamics for each pair of memory spaces. We find that in a pairwise competition, $\mathcal{R}$ dominates both $\mathcal{U}$ and $\mathcal{M}$, whereas $\mathcal{U}$ dominates $\mathcal{M}$. **f,** In a next step, we study the replicator dynamics among all three memory spaces. For the given parameter values, we find that $\mathcal{R}$ is globally stable: independent of the initial composition of the population, all trajectories lead towards a monomorphic population of reactive players. Overall, we obtain a 'memory dilemma': the memory space that evolves is not the memory space that maximizes the population's average payoff.

We note that the final outcome is detrimental to the entire population. Although a homogeneous population of memory-1 players is most cooperative (and hence results in maximum payoffs), such a population is evolutionarily unstable. This type of instability is reflected by the following chain of inequalities:

$$\bar{\pi}_{\mathcal{R}\mathcal{M}} > \bar{\pi}_{\mathcal{M}\mathcal{M}} > \bar{\pi}_{\mathcal{R}\mathcal{R}}. \tag{8}$$

In such a case, we speak of a "memory dilemma". Formally, a memory dilemma arises if the memory space with the largest self payoff (typically the space with the largest memory capacity) can be invaded by a different memory space.

In a next step, we explore how common such memory dilemmas are. To this end, we systematically vary the two main parameters of the model, the selection strength $\beta$ and the cost of cooperation $c$ (Fig 4). This analysis suggests that the occurrence of memory dilemmas is remarkably robust. In particular, for comparably weak selection ($\beta \leq 10$), we find that

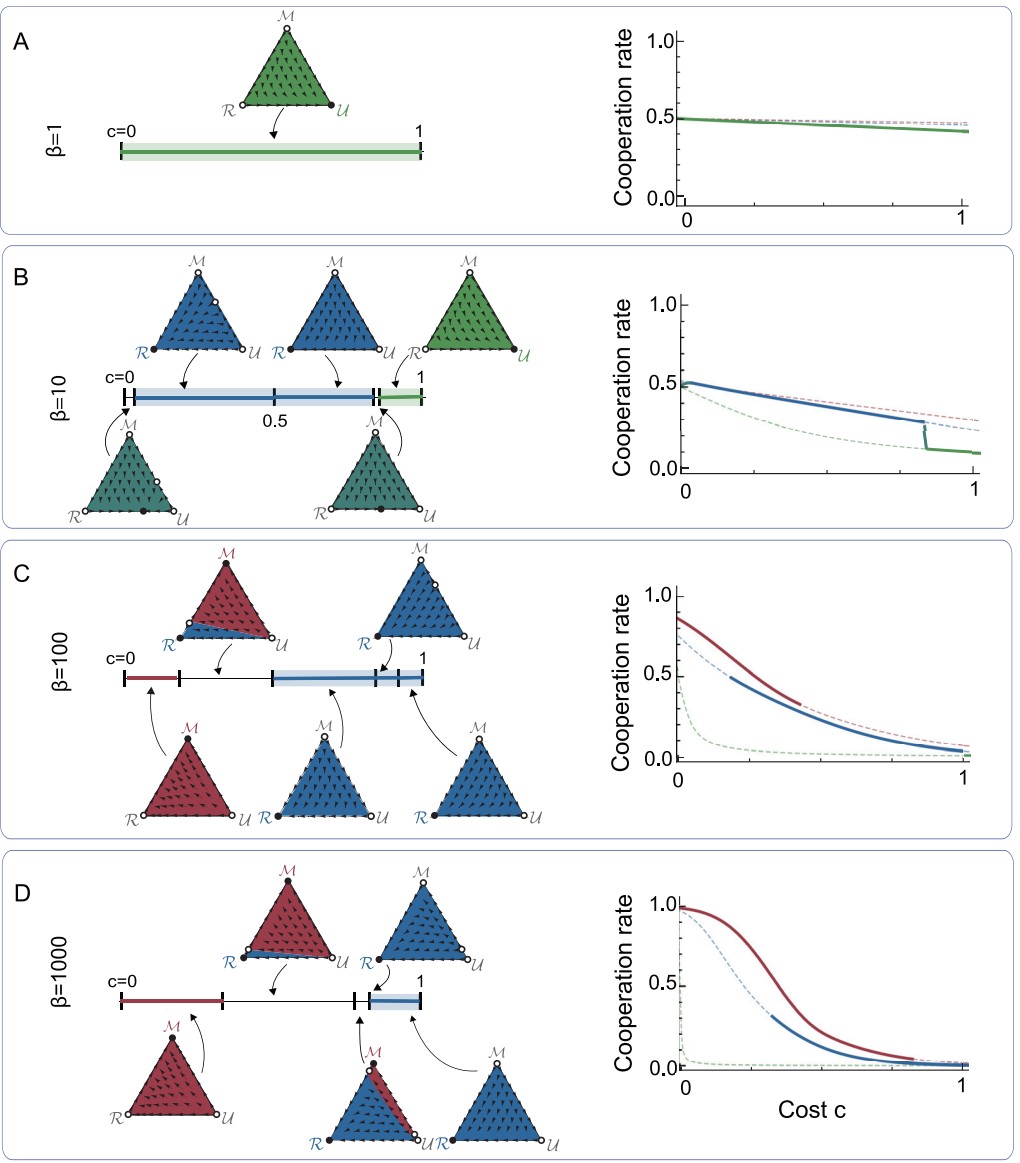

**Fig 4. A bifurcation analysis of the evolutionary dynamics of memory spaces.** We explore the replicator dynamics of the three different memory spaces for four different selection strengths $\beta$. In each case, we first classify the dynamics as we vary the cooperation cost from $c = 0$ to $c = 1$ (left graphs). Here, triangles show representative depictions of the dynamics. Colors indicate basins of attractions of each possible fixed point. If a given memory space $S_i$ is globally stable in a given cost interval, the respective line segment is appropriately colored. Colored shades around these line segments indicate a memory dilemma. Second, we also illustrate the average cooperation rate in monomorphic populations for each of the possible memory spaces (right graphs). If the respective monomorphic population is stable according to replicator dynamics, we use a solid line; otherwise we use a dashed line. **a-d**, For weak selection, unconditional strategies are globally stable for all cost values. As we increase selection strength, memory-1 strategies become stable when cooperation costs are sufficiently small. For large cooperation costs, reactive strategies are globally stable.

memory-1 strategies never evolve, although they typically yield the largest self payoff. In this parameter regime, evolution instead leads to a homogenous population of unconditional players, or of reactive players, or to a mixture of unconditional and reactive players (Fig 4a and 4b). Only when selection is sufficiently strong and when cooperation is cheap, memory-1 players can persist in a population (Fig 4c and 4d). However, even in parameter regimes that allow

for a stable monomorphic population of memory-1 players, inefficiencies might still arise if alternative stable populations exist. For example, for strong selection ($\beta = 1,000$) and an intermediate cost-to-benefit ration ($c/b = 0.4$), both the memory-1 space and the reactive space are locally stable. In that case the initial population determines whether the population eventually adopts the more effective memory-1 space or the less effective reactive space (Fig 4d). The basin of attraction of the reactive equilibrium increases with the cooperation cost $c$. Overall, larger cooperation costs thus increase the likelihood that players settle at an inefficient strategy space.

These results suggest that within the realms of the repeated donation game, memory dilemmas may be surprisingly common. In that case, even if the original dilemma can be avoided when players evolve a sufficient memory capacity, selection pressures may disfavor the evolution of the necessary memory capacities in the first place. We find a qualitatively very similar result when we consider competitions between memory-2 and memory-1 strategies (S7 Fig), suggesting that this finding holds for more complex spaces as well.

To explore how pervasive memory dilemmas are in repeated games, we have extended our analysis to a wide range of other games. In addition to the donation game (and the more general prisoner's dilemma), we also consider the stag-hunt, snowdrift and harmony game (Fig 5 and S10 Fig). We find two key results. First, the memory dilemma extends beyond donation games. In fact, such dilemmas arise frequently when the base game is either a prisoner's dilemma or a snowdrift game. Second, we also find interesting instances of a "reverse" memory dilemma. Here, strategy spaces of lower complexity have the highest self-payoff, yet they can be invaded by more complex strategy spaces. Such cases mostly arise when the underlying game is a harmony game, but they also appear in stag hunt games. Overall, these examples suggest that memory dilemmas can arise easily, in different manifestations and across different types of games.

## Model extensions

Our baseline model focuses on illustrating the fundamental process and the resulting dynamics with a simple setup. We can naturally envision a number of variations of this baseline model. Such variations aid us in exploring the robustness and limits of our results, and give additional insight into the origins of the memory dilemma. In this work, we consider three different main ways of modifying our framework. We present an overview of the effect of these modifications in S8 and S9 Figs, and present tournament results for both strong ($\beta = 100$) and weak selection ($\beta = 10$) in the S1 Data.

In the first major model extension, we explore what happens when we change the method of sampling candidate strategies in the elementary learning process (cf. Eq (5)). For our baseline model, we have assumed that individuals generate new strategies with uniform sampling. That is, when generating a new strategy, each entry $p$ is taken uniformly between $[0, 1]$. As a result, extreme strategies (with values close to 0 and 1) are comparably rare. However, our previous analysis suggests that the memory dilemma arises in part because unconditional and reactive players are more efficient in generating extreme strategies (Fig 2). To explore the effect of extreme strategies in more detail, we may consider an alternative sampling scheme. Instead of uniform sampling, we now assume that a strategy's entries are taken from a U-shaped distribution (the arcsine distribution). However, when we repeat our simulations with this alternative sampling scheme, we observe only a small effect on our results (**Tables K, Q in** S1 Data, S8 (b) and S9(c) Figs). Even when sampling places more weight on extreme behaviors, the lower-memory spaces typically retain (but not expand) their advantage.

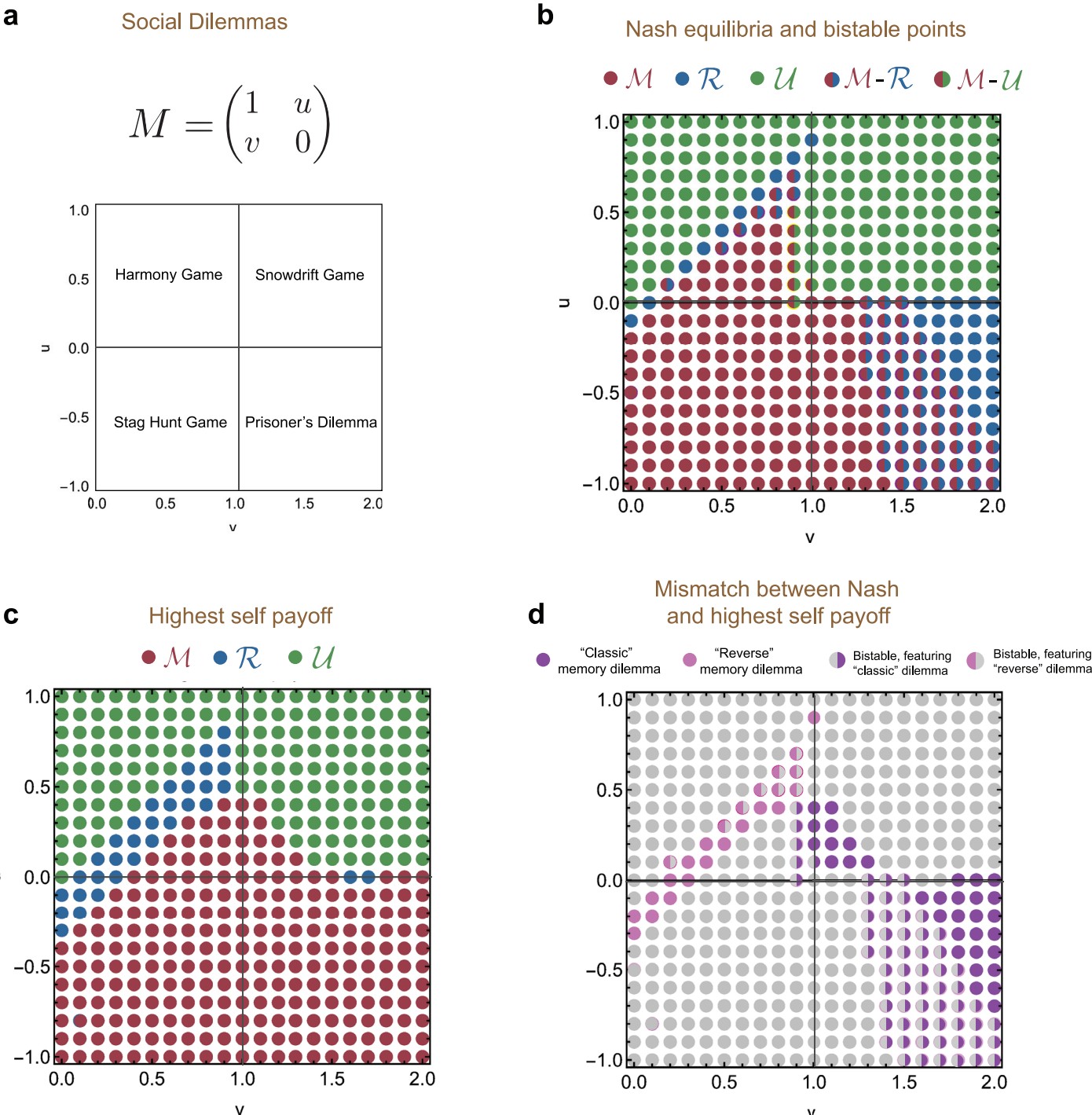

**Fig 5. Evolutionary dynamics of memory spaces for different game structures.** We explore whether memory dilemmas are present in games beyond the repeated Prisoner's Dilemma by repeating our previous simulations for various game matrices. **a**, We consider matrices parametrized by $v$ and $u$. The parameter $v$ varies in the interval $[0, 2]$, and the parameter $u$ varies in $[-1, 1]$. We can partition the resulting two dimensional space of game matrices into four quadrants. The lower right quadrant contains games with a Prisoner's Dilemma (PD) structure like the donation game, whereas the other quadrants contain the other fundamental social dilemmas Snowdrift (SG), Harmony (HG), and Stag Hunt (SH) games. For each of these games, we do the same kind of analysis as for the donation games studied earlier. **b**, First, we depict the Nash equilibria of the $3 \times 3$ payoff matrices when the spaces $\mathcal{M}$, $\mathcal{R}$ and $\mathcal{U}$ compete. For each strategy space, we find parameter regions where this space is an equilibrium. Additionally, we identify regions in which more than one space is stable. **c**, Here, we show the strategy space with the highest self-payoff. Memory-1 strategies tend to get the highest self-payoff in the PD and SH. In the other two game classes, strategy spaces of lower complexity can be more effective. **d**, We distinguish two kinds of memory dilemma. In the "classic" one, the strategy space with highest complexity gives the highest self-payoff, but is not a Nash equilibrium. In the "reverse" one, it is a less complex strategy space that yields the higher self-payoff without being an equilibrium. Both dilemmas also appear in their "weak" forms when bistabilities occur. Parameters: $\beta = 100$, simulations run for $T = 10^9$.

For this extension, we can alternatively envision a different approach to generating new strategies in a way that is more likely to be beneficial for all players. Here, we assume that players search for a new strategy until they have found a better one, instead of only getting one shot per round at revising their strategy. Hence, "bad" strategies are automatically discarded without negatively affecting a player. Yet, we find that this modification only makes the memory dilemma stronger, by increasing the difference between the tournament scores of higher and lower memory (**Tables O, U in** S1 Data). Additionally, self-payoffs of strategy spaces are decreased. It appears that as higher memory players actively search for strategies that are more favorable, they are more likely to only find strategies that are incrementally better. In contrast, if lower memory finds a better strategy, it is still more likely to be more extreme, leading to an even stronger difference in payoffs.

In our second main model extension, we consider a separation of memory capacity and complexity of the strategy space. So far, we have equated a larger memory capacity with higher strategic complexity, interchangably using "memory space" and "strategy space". That is, for our baseline model, we sample one number for unconditional strategies, two numbers for reactive strategies, and four numbers for memory-1 strategies. Given that our previous analysis however hinted at lower memory players having an advantage because of the reduced size of their strategy space, exploring the effect of memory capacity on its own is a relevant research question. We consider two possible methods of disentangling memory capacity from the size of the strategy space by ensuring that the size of the sampled strategy space is independent from the player's memory capacity.

For the first case of this "same-complexity sampling", we sample four random numbers ($p_1$, $p_2$, $p_3$, $p_4$) for each strategy space. Memory-1 strategies are then simply composed of the four sampled values. The two components ($p$, $q$) of reactive strategies are generated by averaging two values $p_i$ each, with $p = (p_1 + p_2)/2$ and $q = (p_3 + p_4)/2$. Finally, the cooperation rate of a player using an unconditional strategy is the average of all four values. When we explore this setup in our simulations, we find that the memory dilemma vanishes for all values of $c$ (S8(d) and S9(e) Figs). Memory-1 players now fully outperform lower memory players (**Tables M, S in** S1 Data). This means that once lower memory players are no longer able to efficiently instantiate extreme behaviors, they lose their ability to win against higher memory strategies. Consequently, this modification resolves the memory dilemma.

For a second variant of same-complexity sampling, we again pick four random numbers for each strategy space. However, we construct reactive strategies of the form ($p$, $q$) in this modification in a different way. For the value of $p$, the player picks the more extreme value out of the two numbers $p_1$ and $p_2$ (i.e., the number that is closer to the boundary of the interval [0, 1]). For the value of $q$, the player takes the more extreme value out of the two numbers $p_3$ and $p_4$. For unconditional strategies, the player chooses the single most extreme value out of all four numbers $p_i$. In contrast to the first method, we observe that this variation still lets lower memory have an advantage (**Tables L, R in** S1 Data), again giving rise to the memory dilemma (S8 (c) and S9(d) Figs). Both methods thus support our previous claim that the memory dilemma arises because the strategy spaces differ in how easily they generate extreme behaviors.

As our third main model extension, we consider an alternative to our introspection dynamics. We have previously used this learning process to describe how individuals update their strategies. Players compare the payoff of their existing strategy to the hypothetical payoff they would have obtained with an alternative strategy. However, this might be a too restrictive choice in some cases. Instead, players may also simply adopt strategies by imitation. In that case, they compare their own payoff to the payoff of the co-player. To allow for such social learning, we assume that players revise their strategies by imitation with probability $\alpha$. With the converse probability ($1 - \alpha$), players use introspection dynamics. Here, one needs to

assume that players are able to infer their co-players' strategies from their observed behaviors. To this end, we assume that higher memory players can directly imitate and adopt lower memory players' strategies. Conversely, when lower memory players imitate a higher memory player, they adopt the low-memory strategy that is most compatible with the higher memory player's behavior. An unconditional player will imitate a memory-1 strategy's average cooperation probability. Similarly, a reactive player will infer effective values of his strategy components $p$ and $q$, which can also be calculated from the invariant distribution [53]. We find that this variation leads to a weakening of the memory dilemma. The payoffs of two players of different memory capacity tend to inch closer together as the imitation probability $\alpha$ increases (**Tables N, T in** S1 Data). As a result, we find more frequent bistability between higher and lower memory (S8(e) and S9(f) Figs).

Overall, we note that memory dilemmas remain present for a range of variations, as long as they do not prevent players of lower memory from coming up with more extreme behaviors than players with higher memory capacity.

## Summary and discussion

Direct reciprocity is an important mechanism for cooperation based on repeated interactions between players. Individuals who engage in direct reciprocity remember the outcomes of previous rounds and use this information to decide their action in the next round. This enables them to conditionally cooperate based on their experiences. How many past interactions a player remembers is governed by the player's memory capacity. Previous studies on direct reciprocity tend to treat a player's memory capacity as fixed and the same for all individuals [14, 16, 35, 54, 55]. Instead, here we consider competitions between players of different memory capacity, i.e. with access to strategy spaces of different complexity. Contrary to what one might expect [18, 21], we find that players with a larger memory are frequently outperformed by lower-memory players. In particular, the evolution of lower memory capacities can easily result in a higher-order social dilemma. Although larger memory capacities tend to lead to higher payoffs in a population, they may be dominated by lower memory.

Existing work on the evolution of memory in games often assumes that memory capacity, or strategic complexity more generally, evolves at a similar time scale as the players' strategies [26, 35]. Instead, here we have considered a setup in which memory capacities evolve at a slower rate. Strategies change and adapt to a present environment due to cultural evolution, which has been suggested to act more rapidly on a society [56, 57]. In contrast, an individual's memory capacity or its ability to deal with cognitive complexity is typically closely tied to the much slower acting mechanism of biological evolution [58, 59].

Our analysis has focused on the simplest three memory spaces, which naturally limits our results. It would be desirable to do the same kind of analysis for more general strategy spaces. However, we note that the complexity of the respective spaces increases dramatically in the number of rounds players can remember. While for memory-2 strategies, some results are still possible (cf. S7 Fig) and show that memory dilemmas exist in this more complex space as well, already the space of memory-3 strategies includes $2^{64} = 1.8 \cdot 10^{19}$ pure (deterministic) strategies. Since our simulations typically introduce between $10^8$ and $10^9$ strategies, a systematic analysis of all memory-3 strategies is already infeasible, even though a calculation of the individual payoffs is possible. We note that there is previous work on simplified strategy spaces containing higher-memory strategies that only count how often players cooperated, but not at which time points [20]. This research, however, also suggests that such simplified strategies are less powerful. For example, they cannot encode behaviors that implicitly depend on the timing of cooperation. Yet, in some instances it may be important to know whether the two players

did in fact cooperate (or defect) in the very same round. Thus, this research suggests that simplified strategies may be less effective in sustaining cooperation overall.

Our results are further restricted by the fact that our findings are based on simulations. While such simulations give interesting insights, an analytical approach could arguably yield a better understanding of the underlying mechanisms. Yet, analytical results in our model would require computing an invariant strategy distribution for each pair of considered strategy spaces. Given that all strategy spaces are continuous and often multidimensional, any suitable analytical approach is mathematically non-trivial.

We note here however that our conclusions are more general than the specific modelling choices might suggest. Recent work [60] has explored a similar question of the competition between memory-1 and reactive strategies without using our two-stage approach to evolution. Instead, players use adaptive optimization strategies by learning from previous actions, which is equivalent to an extension of the (deterministic) coupled replicator dynamics for learning. These dynamics, despite being starkly different to the stochastic introspection dynamics we use in our model, lead to a result that echoes ours: reactive strategies exploit memory-1 strategies and gain the upper hand when both players learn adaptive strategies, leading to a predator-prey relationship. This is shown to happen because memory-1 strategies tend to be more generous towards reactive strategies. This finding is strikingly similar to ours. Hence, this suggests that the observed effect of lower memory winning against higher memory is not driven by the exact choice of dynamics.

For most of our analysis, we focus on the dynamics in simple donation games, which have become one of the main paradigms to study reciprocity [3]. However, our analysis is in no way limited to the particular payoff structure of the donation game. By applying our framework to a wider range of games, we have found that the presence of memory dilemmas is not restricted to the repeated donation game, or even the repeated Prisoner's Dilemma. In fact, memory dilemmas are a more general feature of repeated games, and can arise in different manifestations. For example, they can appear as "classic" memory dilemmas, where more complex strategies can be invaded although they yield higher self payoffs. Alternatively, we have also observed instances of a "reverse" dilemma, where it is the less complex strategies that are unstable despite generating higher self payoffs (Fig 5).

The fact that remembering less information or having limited knowledge in general can be advantageous might not be much of a surprise when we consider literature related to "deliberate ignorance" [61, 62]. When deliberate ignorance is at play, not acquiring or otherwise limiting one's own information can act as a self-commitment or signaling device, which yields a strategic advantage in scenarios such as negotiation or decision making. A similar phenomenon has also been observed in animal behavior, such as in competitions between desert spiders [63]. However, the mechanism behind the advantage of more limited memory in our model is different from self-commitment or signaling: it rather relates to a decreased complexity of the strategy space for lower memory. When the space is less complex, finding a defecting strategy suitable to a hostile environment becomes easier, as a player using this memory space is more flexible in their exploration (see Fig 2).

Importantly, in our model we do not consider the effects of additional complexity costs as in previous work [64, 65]. Such complexity costs may further reinforce the evolution of lower memory spaces. In contrast, we find that larger memory is not disfavored because it is inherently more costly, but because larger memory capacities expand the feasible strategy space. In this way, they make it harder for evolution to "discover" extreme strategies that are most adaptive given the current environment. This aspect is most relevant in hostile environments, in which individuals are incentivized to discover and to adopt defecting strategies. Our results thus highlight the dual nature of memory in direct reciprocity. While memory can be an asset

that allows for sustaining cooperation in favorable conditions, it can also posit a hindrance in environments in which cooperation is unprofitable.

## Methods

To explore the role of memory capacity on cooperation, we consider a model in which the dynamics unfolds on three different levels (S1 Fig). First, there is the game dynamics. Here, two players with given strategies are matched to interact in an infinitely repeated donation game. Second, there is the tournament dynamics. Here two players with given memory capacity are matched, and they are allowed to update their strategies during the tournament. These strategies are constrained by the players' respective memory capacities. Finally, there is the memory dynamics, where also the players' memory capacities are allowed to evolve. In the following, we describe each level in detail.

### Repeated donation game

We consider two players, Player 1 and Player 2, engaging in an infinitely repeated donation game. In each round, players can either cooperate or defect. The focal player's payoff are $b - c$ if both cooperated; $-c$ if only the focal player cooperated and the co-player defected; $b$ if only the focal player defected and the co-player cooperated, and 0 if both defected. We can collect these possible payoffs in the following two vectors,

$$\mathbf{\Pi_1} = (b - c, -c, b, 0)^\mathsf{T} \quad \text{and} \quad \mathbf{\Pi_2} = (b - c, b, -c, 0)^\mathsf{T}. \tag{9}$$

To play these repeated games, players can choose strategies from three possible strategy spaces. These strategy spaces differ in how many past events a player is able to remember. The first option is that a player remembers both players' previous actions. This gives rise to the space of memory-1 strategies

$$\mathcal{M} = \{\ \mathbf{p} = (p_{CC}, p_{CD}, p_{DC}, p_{DD}) \mid 0 \le p_{ij} \le 1\ \}. \tag{10}$$

Here, $p_{ij}$ is the probability that the focal player cooperates in the next round, given the previous actions of the focal player and of the co-player were $i$ and $j$, respectively. The second option is that players remember only the co-player's previous action. Formally, this space of reactive strategies can be identified with the space

$$\mathcal{R} = \{\ \mathbf{p} \in \mathcal{M} \mid p_{CC} = p_{DC},\ p_{DC} = p_{DD}\ \}. \tag{11}$$

Here, $p_{XC} := p_{CC} = p_{DC}$ is the player's probability to cooperate, given the co-player cooperated in the previous round. Finally, players may remember no previous events. In this case, they are restricted to use unconditional strategies, which can be identified with the set

$$\mathcal{U} = \{\ \mathbf{p} \in \mathcal{M} \mid p_{CC} = p_{DC} = p_{DC} = p_{DD}\ \}. \tag{12}$$

Here, $p_{XC} := p_{CC} = p_{DC}$ is the player's probability to cooperate in any given round. Throughout the text, we refer to these three sets as either the players' 'strategy space' or as their 'memory space'.

 If both players use a strategy in $\mathcal{M}$ (or one of its subsets), their payoffs can be computed explicitly [2]. To this end, suppose Player 1 uses strategy $\mathbf{p} = (p_{CC}, p_{CD}, p_{DC}, p_{DD})$ and Player 2 uses strategy $\mathbf{q} = q_{CC}, q_{CD}, q_{DC}, q_{DD})$. Then the repeated donation game can be described as a Markov chain. The possible states of this Markov chain are the possible outcomes in each round, *CC, CD, DC, DD*, where the first letter refers to player 1's action and the second letter

to player 2's action. The transition matrix of this Markov chain is given by

$$
\begin{array}{c}
\phantom{CC} \quad CC \qquad\quad CD \qquad\qquad DC \qquad\qquad DD \\
\begin{array}{c} CC \\ CD \\ DC \\ DD \end{array}
\begin{pmatrix}
p_{CC}q_{CC} & p_{CC}(1-q_{CC}) & (1-p_{CC})q_{CC} & (1-p_{CC})(1-q_{CC}) \\
p_{CD}q_{DC} & p_{CD}(1-q_{DC}) & (1-p_{CD})q_{DC} & (1-p_{CD})(1-q_{DC}) \\
p_{DC}q_{CD} & p_{DC}(1-q_{CD}) & (1-p_{DC})q_{CD} & (1-p_{DC})(1-q_{CD}) \\
p_{DD}q_{DD} & p_{DD}(1-q_{DD}) & (1-p_{DD})q_{DD} & (1-p_{DD})(1-q_{DD})
\end{pmatrix}.
\end{array}
\tag{13}
$$

Assuming that the transition matrix is primitive (which is true, for example, if the players' actions are subject to rare implementation errors, such that strategies are never fully deterministic), this Markov chain converges to a unique invariant distribution, $\mathbf{v} = (v_{CC}, v_{CD}, v_{DC}, v_{DD})$. Given this invariant distribution, we can define the players' payoffs as follows

$$
\pi_1 = \mathbf{v} \cdot \mathbf{\Pi_1} \quad \text{and} \quad \pi_2 = \mathbf{v} \cdot \mathbf{\Pi_2}.
\tag{14}
$$

We note that the payoffs of more general memory$-k$ strategies can be computed with analogous Markov chain methods. However, while the respective payoffs can be easily calculated, a systematic analysis of the respective strategies is out of reach, due to the enormous size of these strategy spaces.

Alternatively, we can also compute payoffs using the formula of Press and Dyson [26],

$$
\pi_1 = \frac{D(\mathbf{p}, \mathbf{q}, \mathbf{\Pi_1})}{D(\mathbf{p}, \mathbf{q}, \mathbf{1})} \quad \text{and} \quad \pi_2 = \frac{D(\mathbf{p}, \mathbf{q}, \mathbf{\Pi_2})}{D(\mathbf{p}, \mathbf{q}, \mathbf{1})}.
\tag{15}
$$

For an arbitrary 4-dimensional vector $\mathbf{f} := (f_{CC}, f_{CD}, f_{DC}, f_{DD})$, the term $D(\mathbf{p}, \mathbf{q}, \mathbf{f})$ is defined as

$$
D(\mathbf{p}, \mathbf{q}, \mathbf{f}) = \det \begin{pmatrix}
-1 + p_{CC}q_{CC} & -1 + p_{CC} & -1 + q_{CC} & f_{CC} \\
p_{CC}q_{DC} & -1 + p_{CD} & q_{DC} & f_{CD} \\
p_{DC}q_{CD} & p_{DC} & -1 + q_{CD} & f_{DC} \\
p_{DD}q_{DD} & p_{DD} & q_{DD} & f_{DD}
\end{pmatrix}.
\tag{16}
$$

This formalism allows us to compute the players' payoffs, for any generic strategies $\mathbf{p}, \mathbf{q} \in \{\mathcal{M}, \mathcal{R}, \mathcal{U}\}$.

## Tournaments

As a first approach to study games among players with different memory capacities, we consider tournaments. To this end, we suppose the two players' memory capacities are fixed. As a result, player 1 can choose among all strategies in the respective memory space $\mathcal{S}_1$ and player 2 can choose among all strategies in $\mathcal{S}_2$, with $\mathcal{S}_1, \mathcal{S}_2 \in \{\mathcal{M}, \mathcal{R}, \mathcal{U}\}$. During these tournaments, players do not play fixed strategies; rather they learn to adopt new strategies over time while adapting to their opponent. Here, we note that our study aims to explore what happens when there are differences between players in terms of their cognitive abilities. In contrast, an implicit assumption in evolutionary game theory is often that games are symmetric. Players have the same strategic options and the same things they can remember. In such cases, players can learn new strategies for example by imitation. Learning processes for players in asymmetric games are less straightforward to model. Here, we use one particular model, introspection dynamics, which has been used previously to analyze learning in repeated social dilemmas [48,

49]. Based on this learning dynamics, we define the payoff $\bar{\pi}_{\mathcal{S}_1 \mathcal{S}_2}$ of a player with memory space $\mathcal{S}_1$ against a player with memory space $\mathcal{S}_2$ by Eq (6).

For the tournament, we compute $\bar{\pi}_{\mathcal{S}_1 \mathcal{S}_2}$ for all possible combinations $\mathcal{S}_1, \mathcal{S}_2 \in \{\mathcal{M}, \mathcal{R}, \mathcal{U}\}$ by simulating the introspection dynamics for $T = 10^9$ time steps. We use four different measures $\varphi(\mathcal{S}_i)$ to quantify the overall success of a given memory space $\mathcal{S}_i$ (Tables 1–4, **A–I in** S1 Data):

1. The number of (pairwise) wins,

$$\varphi(\mathcal{S}_i) = \sum_{\mathcal{S}_j \neq \mathcal{S}_i} \mathbf{1}_{\bar{\pi}_{\mathcal{S}_i, \mathcal{S}_j} \geq \bar{\pi}_{\mathcal{S}_j, \mathcal{S}_i}}. \tag{17}$$

Here, $\mathbf{1}_x$ is an indicator function. It equals one if the statement $x$ is true; it equals zero otherwise.

2. The score,

$$\varphi(\mathcal{S}_i) = \sum_{\mathcal{S}_j \neq \mathcal{S}_i} \bar{\pi}_{\mathcal{S}_i, \mathcal{S}_j}. \tag{18}$$

3. The memory space's self score,

$$\varphi(\mathcal{S}_i) = \bar{\pi}_{\mathcal{S}_i, \mathcal{S}_i}. \tag{19}$$

4. The combined score,

$$\varphi(\mathcal{S}_i) = \sum_{\mathcal{S}_j} \bar{\pi}_{\mathcal{S}_i, \mathcal{S}_j}. \tag{20}$$

## Evolutionary dynamics of memory spaces

In addition to the previous static analysis, we have also considered a dynamics in which the players' memory spaces are a biological trait that can evolve in time. To this end, we consider an infinite population of players. Each player is equipped with a given memory space. Let $x_{\mathcal{M}}$, $x_{\mathcal{R}}$, and $x_{\mathcal{U}}$ denote the respective proportions with which each memory space is used in the population. Players are randomly matched to engage in pairwise donation games with fixed parameters $c$ and $\beta$. If a player with memory space $\mathcal{S}_i$ interacts with a co-player with space $\mathcal{S}_j$, the player's payoffs are $\bar{\pi}_{\mathcal{S}_i \mathcal{S}_j}$ and $\bar{\pi}_{\mathcal{S}_j \mathcal{S}_i}$, as defined by Eq (6). For the three memory-spaces considered herein, we can assemble these pairwise payoffs in a 3×3 payoff matrix $M$

$$
\begin{array}{c}
 \\
\mathcal{M} \\
\mathcal{R} \\
\mathcal{U}
\end{array}
\begin{array}{ccc}
\mathcal{M} & \mathcal{R} & \mathcal{U}
\end{array}
\left(
\begin{array}{ccc}
\bar{\pi}_{\mathcal{M}\mathcal{M}} & \bar{\pi}_{\mathcal{M}\mathcal{R}} & \bar{\pi}_{\mathcal{M}\mathcal{U}} \\
\bar{\pi}_{\mathcal{R}\mathcal{M}} & \bar{\pi}_{\mathcal{R}\mathcal{R}} & \bar{\pi}_{\mathcal{R}\mathcal{U}} \\
\bar{\pi}_{\mathcal{U}\mathcal{M}} & \bar{\pi}_{\mathcal{U}\mathcal{R}} & \bar{\pi}_{\mathcal{U}\mathcal{U}}
\end{array}
\right), \tag{21}
$$

For $c \in \{0.1, 0.2\ldots, 0.9\}$ and $\beta = \{1, 10, 100, 1000\}$, the respective entries of this matrix can be taken from the tables **Tab. 1–4, A–I in** S1 Data. Eq (7) displays an example of this matrix for

$c = 0.6$ and $\beta = 10$. Similar matrices can be created for any value of $c$ and $\beta$; for example, for Fig 4 that shows the bifurcations of the system, we have created $c$ values using a step size of 0.005.

We assume that memory spaces that yield a high payoff become more abundant in the population. Many mathematical models could be used to describe the evolution of strategy spaces; here, we use the replicator equation [50] to model these long-run dynamics. However, many of our results are independent of the specific dynamics we consider. For example, if one memory space is strictly dominated, then the abundance of players who use that strategy space is expected to decrease for any payoff-monotone dynamics. This includes replicator dynamics but also many others [51].

The fitness of a trait is proportional to the payoff it gets in the population. That is, the fitness of memory space $\mathcal{S}_i$ is

$$f_{\mathcal{S}_i} = \sum_{\mathcal{S}_j} \bar{\pi}_{\mathcal{S}_i \mathcal{S}_j} \cdot x_{\mathcal{S}_j}. \tag{22}$$

The average fitness in the population $\bar{f}$ is then given by

$$\bar{f} = \sum_{\mathcal{S}_i} f_{\mathcal{S}_i} \cdot x_{\mathcal{S}_i}. \tag{23}$$

Based on these two terms, replicator dynamics posits that the proportions of the three memory spaces change according to the ordinary differential equation

$$\dot{x}_{\mathcal{S}_i} = x_{\mathcal{S}_i} \left( f_{\mathcal{S}_i} - \bar{f} \right) \tag{24}$$

This replicator equations is defined on the unit simplex $S_3$, which is described by $x_{\mathcal{M}} + x_{\mathcal{R}} + x_{\mathcal{U}} = 1$. This unit simplex is represented by triangles in Figs 3 and 4. The corners of this triangle correspond to homogeneous populations, where $x_{\mathcal{S}_i} = 1$ for some $\mathcal{S}_i$. The edges of the triangle correspond to all populations where one of the three memory spaces is absent, $x_{\mathcal{S}_i} = 0$ for some $\mathcal{S}_i$. Finally, the interior of the simplex corresponds to all population mixtures where each memory space is adopted by a positive proportion of players. We note that faces of the simplex as well as the interior are invariant: a strategy that is not there initially will not appear, and trajectories that start inside the simplex will not reach the boundary in finite time (although they may converge to it). All vertices of the simplex are rest points/stationary solutions of the dynamics. They can be stable or unstable (see Fig 3a). To illustrate these dynamics, we use the Dynamo 3S package for Mathematica [66].

In Fig 3, we show an example of the resulting replicator dynamics, for a particular value of $c = 0.6$ and $\beta = 10$. In Fig 4, we characterize all possible dynamics that occur in our system, for four different selection strengths, $\beta \in \{1, 10, 100, 1000\}$ and $c \in (0, 1)$. This figure is complemented by S2 Fig, which illustrates all bifurcations that occur on the boundary of the state space. Finally, S3 and S4 Figs further generalize these results to cases in which cooperation is inefficient ($c > b = 1$), and to cases in which cooperation is individually profitable ($c < 0$).

## Supporting information

**S1 Fig. The different conceptual levels of our analysis.** We consider dynamics on three different levels. **a**, The most elementary level is the basic repeated donation game. On this level, we consider two players with fixed strategies **p** and **q**. These strategies are either memory-1 ($\mathcal{M}$), reactive ($\mathcal{R}$), or unconditional ($\mathcal{U}$). Given the strategies, we can compute the players' payoffs by either simulating the game dynamics, or by using a Markov chain approach. **b**, On the next level, we conduct tournaments among players with fixed memory capacity. Here,

players interact in many pairwise repeated games with the same opponent. Between each repeated game, players are permitted to update their strategies (subject to their respective memory constraint). Here, we illustrate the dynamics between a reactive player and an unconditional player. The space of reactive strategies is 2-dimensional (represented by a square). The space of unconditional strategies is 1-dimensional (represented by a line segment). The overall state space can thus be represented by a cube. Dots within this cube represent the players' strategies at any point of the learning process. By running this learning process for sufficiently long, we can compute the expected payoff of a reactive player against an unconditional player. **c**, Based on these payoffs, we also explore the evolutionary dynamics when the players' strategy spaces themselves are subject to evolution. We describe this strategy space dynamics with the replicator equation. For details, see Methods.
(EPS)

**S2 Fig. A pairwise bifurcation analysis of the evolutionary dynamics of strategy spaces.** This figure builds on Fig 4, by clarifying the dynamics on the three boundaries of the unit simplex. That is, for each selection strength $\beta = \{1, 10, 100, 1000\}$ and for all cost values $c \in (0, 1)$ we show the bifurcations that occur between each pair of strategy spaces, $\mathcal{R}$ vs $\mathcal{U}$, $\mathcal{M}$ vs $\mathcal{R}$, and $\mathcal{M}$ vs $\mathcal{U}$.
(EPS)

**S3 Fig. The dynamics of strategy spaces beyond the social dilemma case.** Here we extend the analysis shown in Fig 4 by also allowing for cases in which cooperation is individually profitable ($c < 0$), or when cooperation is inefficient ($c > b = 1$). In both cases, we find for weak selection that unconditional strategies are globally stable (indicated by green colors). In contrast, for strong selection, memory-1 strategies (red) are dominant when cooperation is individually profitable, whereas unconditional strategies remain dominant when cooperation is inefficient.
(EPS)

**S4 Fig. Resulting cooperation rates when the players' memory capacities are subject to evolution.** This figure complements the analysis in Fig 4 by showing the possible cooperation rates for all $c \in (-1, 2)$. For strong selection, we observe that all memory spaces yield almost full cooperation for $c < 0$, whereas all memory spaces result in almost universal defection when $c > b = 1$, as one may expect.
(EPS)

**S5 Fig. Joint distribution of cooperation rates among memory-1 and reactive players.** Similar to Fig 2, this figure illustrates the players' cooperation rates either at their first encounter ($t = 0$), or after a few learning steps have taken place ($t = 10$). Here we consider three scenarios: both players are reactive; both players are memory-1; or one player is reactive, the other one is memory-1. Based on this figure, we can make two observations. First, reactive players are more likely to adopt strategies with extreme cooperation rates. This can be seen, for example, in panel **c**, where the cooperation rate of memory-1 players is more closely centered around 50%. Second, in environments in which defection is profitable, reactive players are quicker to reduce their cooperation rates. This can be seen in panel **f**. Here, pairs of players are more likely to be above the diagonal, where the memory-1 player is more cooperative than the reactive player. Parameters are the same as in Fig 2.
(EPS)

**S6 Fig. Joint distribution of cooperation rates among memory-1 and unconditional players.** This figure shows the same type of result as the previous figure, but this time illustrating

the dynamics among memory-1 players and unconditional players. The qualitative results are similar to the qualitative results of S5 Fig.
(EPS)

**S7 Fig. Lower memory outcompetes higher memory also when we allow for memory-2 strategies. a–c**, For a fixed value of $c = 0.5$, the average payoff for a memory-1 player (red) against another one of their kind evolves to a lower value (**a**) than the payoff of a memory-2 player (pink) playing against another memory-2 player (**b**). On the other hand, payoff averages of memory-1 strategies playing against memory-2 strategies quickly evolve to give the lower memory player a clear advantage (**c**) after not even 100 timesteps. Averages are taken over 1000 runs. **d**, We observe the advantage of memory-1 strategies for values of cost $c$ going from $c = 0.1$ to $c = 1$, with simulations running for $10^6$ timesteps. Parameters: $\beta = 10$, $b = 1$.
(EPS)

**S8 Fig. Model extensions.** We consider various variations of the baseline model at a fixed cost value of $c = 0.5$, under strong selection. **a**, In our original baseline model, we find a memory dilemma for these parameter values: the space of reactive strategies $\mathcal{R}$ is the Nash equilibrium, whereas $\mathcal{M}$ gains the highest self-payoff. **b**, When we sample new strategies from a U-shaped distribution instead of a uniform distribution, we again find the same memory dilemma. Self-payoffs are however increased for both $\mathcal{M}$ and $\mathcal{R}$. **c**, We consider same-complexity sampling with biasing, where we draw four random values no matter the strategy space, and then construct lower memory strategies by picking the values that are closest to the boundary. To compose reactive strategies, we pick the two most extreme values, and for unconditional strategies, we pick the one most extreme value. We again find the same memory dilemma as in the baseline model. Self-payoffs of $\mathcal{R}$ and $\mathcal{U}$ are increased. **d**, We again consider same-complexity sampling that draws four random values no matter the strategy space. For this extension, we however construct lower memory strategies by averaging two values each (for reactive strategies) or averaging all four values (for unconditional strategies). Now, the memory dilemma vanishes: $\mathcal{M}$ both gains the highest self-payoff and is the Nash equilibrium. **e**, Players can imitate the opponent's strategy with probability $\alpha = 0.05$. Lower-memory players infer the strategies of higher memory players, which can be calculated via the invariant distribution arising from the repeated game. Here, we find a "weakened" memory dilemma: $\mathcal{M}$ and $\mathcal{R}$ are bistable, while $\mathcal{M}$ has the highest self-payoff. Parameters: $\beta = 100$, $b = 1$, simulations run for $T = 10^9$.
(EPS)

**S9 Fig. The effect of varying the baseline model on the evolutionary dynamics of memory spaces. a**, We consider three levels of classification: the strategy space with the highest self-payoff, the strategy space that is the Nash equilibrium or the spaces that are bistable, and the presence of a memory dilemma, indicated by the Nash equilibrium not matching the highest self-payoff. We distinguish two kinds of memory dilemma. One is the "classic" memory dilemma, where higher memory has a higher self-payoff, whereas it is not a Nash equilibrium. The other is the "reverse" memory dilemma, characterized by lower memory having the highest self-payoff, and higher memory being the Nash equilibrium instead. **b**, In the baseline model, we find results in accordance with Fig 4: for intermediate values of $c$, there is a "weak" memory dilemma in the form of a bistability between $\mathcal{M}$ and $\mathcal{R}$, whereas for increasing cost, the memory dilemma becomes stronger, with $\mathcal{R}$ being the Nash equilibrium, while $\mathcal{M}$ has the highest self-payoff. **c**, U-shaped distribution sampling results in very similar dynamics as the baseline model. For very low values of $c$, we see that $\mathcal{M}$ can keep its advantage a bit better. **d**, When we do same-complexity sampling with biasing for lower memory strategies, the picture also remains similar in comparison to the baseline. **e**, For same-complexity sampling with

averaging, we see the memory dilemma vanish for every value of $c$. $\mathcal{M}$ has the highest self-payoff and also is the Nash equilibrium throughout the entire range of cost values. **f,** When players can imitate others' strategies with probability $\alpha = 0.05$ either by direct copying or inference, the memory dilemma is weakened. This means that for a wide range of $c$-values, $\mathcal{M}$ and $\mathcal{R}$ are bistable while $\mathcal{M}$ has the highest self-payoff. Parameters as in the previous figure.
(EPS)

**S10 Fig. Evolutionary dynamics of memory spaces for different game structures: Weak selection.** We present a figure analogous to Fig 5, this time for $\beta = 10$. Here, a richer variety of dynamics can unfold. Here, memory dilemmas appear in almost the entire quadrant containing Prisoner's Dilemma matrices: either $\mathcal{R}$ is the Nash equilibrium, or a mixture of $\mathcal{R}$ and $\mathcal{U}$ is stable. We also see memory dilemmas in some Snowdrift games, as well as Stag Hunt games. On the other hand, reverse dilemmas only show up in their weak form, with a coexistence of $\mathcal{R}$ and $\mathcal{U}$. Parameters: $\beta = 10$, simulations run for $T = 10^9$.
(EPS)

**S1 Data. Tournament results for different selection strengths and model extensions.**
**Table A. Pairwise competitions under very weak selection**. The table shows the same type of data as Table 1, but using a selection strength of $\beta = 1$ instead of $\beta = 100$. We find that lower memory always wins in a pairwise competition, regardless of the value of $c$. Simulations are run for $T = 10^9$ time steps. **Table B. Wins and scores under very weak selection**. The table shows the same type of data as Table 2, but again using a selection strength of $\beta = 1$ instead of $\beta = 100$. With respect to both measures, wins and scores, $\mathcal{U}$ is always ranked first. Simulations are run for $T = 10^9$ time steps. **Table C. Self scores and combined scores under very weak selection**. The table shows the same type of data as Table 3, but for $\beta = 1$ instead of $\beta = 100$. The space of memory-1 strategies tends to have the largest self-payoff, but unconditional strategies have the largest combined score across all cost values. Simulations are run for $T = 10^9$ time steps. **Table D. Pairwise competitions under rather weak selection**. The table shows the same type of data as Table 1, but using a selection strength of $\beta = 10$. Similar to the case of very weak selection, we find that lower memory always wins in a pairwise competition. Simulations are run for $T = 10^9$ time steps. **Table E. Wins and scores under rather weak selection**. The table shows the same type of data as Table 2, but using a selection strength of $\beta = 10$. Similar to the case of very weak selection, $\mathcal{U}$ is always ranked first. Simulations are run for $T = 10^9$ time steps. **Table F. Self scores and combined scores under rather weak selection**. The table shows the same type of data as Table 3, but with $\beta = 10$. Again, the space of memory-1 strategies tends to have the largest self-payoff, but reactive and unconditional strategies have the largest combined score. Simulations are run for $T = 10^9$ time steps. **Table G. Pairwise competitions under very strong selection**. The table shows the same type of data as Table 1, but using a selection strength of $\beta = 1000$. Reactive players always win against unconditional players. They also win against memory-1 players when $c \geq 0.2$. Simulations are run for $T = 10^9$ time steps. **Table H. Wins and scores under very strong selection**. The table shows the same type of data as Table 2, but using a selection strength of $\beta = 1000$. In most cases, reactive strategies rank first with respect to both, wins and score. Simulations are run for $T = 10^9$ time steps. **Table I. Self scores and combined scores under very strong selection**. The table shows the same type of data as Table 3, but with $\beta = 1000$. Memory-1 strategies have the largest self payoff. They also have the largest combined score provided $c \leq 0.7$. Simulations are run for $T = 10^9$ time steps. **Table J. Self scores and combined scores in pairwise tournaments**. For ease of comparison with the data below, we present the data from Table 3 once more. Across all cooperation costs, we find that memory-1 strategies yield the largest self payoff. They also achieve the largest combined score if $c \leq 0.3$; otherwise, for $c \geq 0.4$, reactive strategies succeed.

Simulations are run for $T = 10^9$ time steps. **Table K. Self scores and combined scores for Extension 1 under** $\beta = 100$. The table shows the same type of data as Table 3, but for the model extension where we sample all strategies from a U-shaped distribution. We find that memory-1 strategies retain the largest self payoff for each value of $c$. They also have the largest combined score for $c < 0.4$, whereas reactive strategies win for higher values of $c$. Simulations are run for $T = 10^9$ time steps. **Table L. Self scores and combined scores for Extension 2 under** $\beta = 100$. The table shows the same type of data as Table 3, but for the model extension where we sample strategies by drawing four values each, and then choosing the value(s) closest to the boundary for reactive and unconditional strategies. This biases strategies away from 0.5. We find that memory-1 strategies retain the largest self payoff for each value of $c$. They also have the largest combined score for $c < 0.3$, whereas reactive strategies win for higher values of $c$. Simulations are run for $T = 10^9$ time steps. **Table M. Self scores and combined scores for Extension 3 under** $\beta = 100$. The table shows the same type of data as Table 3, but for the model extension where we sample four values for each strategy space, and average them to construct lower memory strategies. Reactive strategies are composed by averaging two values each to get a tuple, whereas unconditional strategies are constructed by averaging all four sampled values to get the player's cooperation probability. Memory-1 strategies have the largest self payoff. They also have the largest combined score for all values of $c$. Simulations are run for $T = 10^9$ time steps. **Table N. Self scores and combined scores for Extension 4 under** $\beta = 100$. The table shows the same type of data as Table 3, but for the model extension where high memory players imitate their co-player's strategy with probability $\alpha = 0.05$, whereas with the same probability low-memory players infer their co-player's effective memory strategy. Memory-1 strategies still have the largest self payoff. They also have the largest combined score for $c > 0.4$, whereas reactive strategies win for higher values of $c$. We note however that compared to Table 3, the competition is tighter. Simulations are run for $T = 10^9$ time steps. **Table O. Self scores and combined scores when players can actively search for better strategies under** $\beta = 100$. The table shows the same type of data as Table 3, but for the model extension where players are allowed to search for a new strategy until the mutant is accepted. We find that Memory-1 strategies still have the largest self payoff. They also have the largest combined score for $c < 0.2$, whereas reactive strategies win for higher values of $c$. Simulations are run for $T = 10^9$ time steps. **Table P. Self scores and combined scores under rather weak selection**. The table shows the same type of data as Table 3, but with $\beta = 10$. Again, the space of memory-1 strategies tends to have the largest self-payoff, but reactive and unconditional strategies have the largest combined score. Simulations are run for $T = 10^9$ time steps. **Table Q. Self scores and combined scores for Extension 1 under** $\beta = 10$. The table shows the same type of data as Table 3, but for the model extension where we sample all strategies from a U-shaped distribution. We find that memory-1 strategies retain the largest self payoff for each value of $c$. Meanwhile, reactive strategies have the largest combined score for all values of $c$. Simulations are run for $T = 10^9$ time steps. **Table R. Self scores and combined scores for Extension 2 under** $\beta = 10$. The table shows the same type of data as Table 3, but for the model extension where we sample strategies by drawing four values each, and then choosing the value(s) closest to the boundary for reactive and unconditional strategies. This biases strategies away from 0.5. We find that memory-1 strategies retain the largest self payoff for $c > 0.1$. However, reactive strategies have the largest combined score for $c < 0.8$, whereas unconditional strategies win for higher values of $c$. Simulations are run for $T = 10^9$ time steps. **Table S. Self scores and combined scores for Extension 3 under** $\beta = 10$. The table shows the same type of data as Table 3, but for the model extension where we sample four values for each strategy space, and use averaging to construct lower memory strategies. Reactive strategies have the largest self payoff for most values of $c$. However, memory-1 has the largest combined score for all values of $c$.

Simulations are run for $T = 10^9$ time steps. **Table T. Self scores and combined scores for Extension 4 under $\beta$ = 10.** The table shows the same type of data as Table 3, but for the model extension where high memory players imitate their co-player's strategy with probability $\alpha$ = 0.05, whereas with the same probability low-memory players infer their co-player's effective memory strategy. Memory-1 strategies still have the largest self payoff. However, reactive strategies have the largest combined score for $c < 0.9$. Simulations are run for $T = 10^9$ time steps. **Table U. Self scores and combined scores for Extension 5 under $\beta$ = 10.** The table shows the same type of data as Table 3, but for the model extension where players are allowed to search for a new strategy until the mutant is accepted. Memory-1 strategies still have the largest self payoff except for $c = 0.1$. However, unconditional strategies have the largest combined score for all values of $c$. Simulations are run for $T = 10^9$ time steps.
(PDF)

## Author Contributions

**Conceptualization:** Laura Schmid, Christian Hilbe, Krishnendu Chatterjee, Martin A. Nowak.

**Data curation:** Laura Schmid, Martin A. Nowak.

**Formal analysis:** Laura Schmid, Christian Hilbe, Martin A. Nowak.

**Funding acquisition:** Christian Hilbe, Krishnendu Chatterjee.

**Investigation:** Laura Schmid, Christian Hilbe, Martin A. Nowak.

**Methodology:** Christian Hilbe, Krishnendu Chatterjee, Martin A. Nowak.

**Project administration:** Krishnendu Chatterjee, Martin A. Nowak.

**Software:** Martin A. Nowak.

**Supervision:** Christian Hilbe, Krishnendu Chatterjee, Martin A. Nowak.

**Validation:** Laura Schmid, Christian Hilbe, Martin A. Nowak.

**Visualization:** Laura Schmid.

**Writing – original draft:** Laura Schmid.

**Writing – review & editing:** Laura Schmid, Christian Hilbe, Martin A. Nowak.

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
