## [Decision Letter · Decision Letter 0]

3 Feb 2022

Dear Ms Schmid,

Thank you very much for submitting your manuscript "Direct reciprocity between individuals that use different strategy spaces" for consideration at PLOS Computational Biology.

As with all papers reviewed by the journal, your manuscript was reviewed by members of the editorial board and by several independent reviewers. In light of the reviews (below this email), we would like to invite the resubmission of a significantly-revised version that takes into account the reviewers' comments.

Your manuscript has been evaluated by 3 reviewers with expertise in, respectively, evolutionary biology, game theory, and cognitive and social psychology. While the reviewers find your work relevant and interesting, they ask for better clarification of the game and model, and the exploration or discussion of alternative possibilities.

We cannot make any decision about publication until we have seen the revised manuscript and your response to the reviewers' comments. Your revised manuscript is also likely to be sent to reviewers for further evaluation.

Sincerely,

Lusha Zhu, Ph.D.

Associate Editor

PLOS Computational Biology

James O'Dwyer

Deputy Editor

PLOS Computational Biology

Your manuscript has been evaluated by 3 reviewers with expertise in, respectively, evolutionary biology, game theory, and cognitive and social psychology. While the reviewers find your work relevant and interesting, they ask for better clarification of the game and model, and the exploration or discussion of alternative possibilities.

Reviewer's Responses to Questions

**Comments to the Authors:**

Reviewer #1: Schmid and colleagues studied repeated interactions between agents with different strategy spaces that required different memory capacity: unconditional players who have no memory, reactive players who responded to the last move of the co-player, and memory-1 (M1) players who consider the last moves of both parties. Combining numerical calculation and simulation, they conducted careful comparisons on the dominance of different types of agents and examined the dynamics at two scales: (i) when the strategy space of the agents was fixed but the strategies were allowed to evolve (tournaments); (ii) when the strategy space also evolved across time based on the payoff of each type of strategies (evolutionary dynamics). Their results point to an interesting “memory dilemma”: the society will be better off if all players adopt M1 strategies, while the sophisticated strategies can only dominate more naive ones under few conditions.

Overall, the research question is important, the analyses are carefully conducted, and the paper is clearly written. However, in my opinion, the memory dilemma demonstrated by the authors would have more significant implications if they provide more results or discussions on the specificity, robustness, or generality of the memory dilemma. Although the authors have already discussed the commonality of the memory dilemma in the parameter space for both tournaments and evolutionary dynamics, these analyses were based on the same assumptions including how agents change their strategies within their corresponding spaces.

Specifically, it is not clear whether the failure of M1 players is caused by the memory capacity itself (i.e., the sophistication of strategies or the contingency of strategies on the interactive histories) or by the inefficient search of the strategy space (associated with the size of the strategy space or the efficiency of the search algorithm) given the assumptions of how strategies evolve. This alternative explanation is consistent with the speculation of the authors about the possible reason why M1 strategies fails –– it is relatively hard for M1 players to come up with extreme behavior, which prevent them from switching quickly to defect when cooperation is costly or the selection is weak. I thus have three main questions:

1. Does the failure specific to strategy space that requires higher memory capacity or general to strategy space that is large in size? Alternatively, do reactive/unconditional players still outperform M1 players if the sizes of their strategy spaces are matched? Although higher memory capacity often leads to larger size of the strategy space, these two factors can still be dissociated. For example, given the memory capacity, the size of the strategy space can still vary (e.g., by sampling different numbers of strategies to construct subsets of the entire space).

2. What if agents are allowed to use other algorithms that are potentially more efficient to search the strategy space, compared with the current algorithm that randomly select a new strategy and adopt it with a certain probability based on the payoffs? For example, an algorithm that reduces the probability of sampling from a local sub-space, if a strategy in it generates a very low payoff.

3. Social learning is also an important way to improve the search in the strategy space. Although players with lower memory capacity cannot imitate strategies of those with higher memory capacity, it is possible for the latter to imitate the former. Do the memory dilemma still hold if M1 players imitate others’ strategies or simply copy others’ actions when others’ payoffs are higher than themselves?

Minor points:

1. How many times of simulation were performed for the tournaments? The authors should probably specify this in the Methods part if they have not.

2. The authors could expand their results or discussions a bit on strategies that require higher memory capacity. Although the strategy space grows dramatically with the memory capacity, using heuristics may help reduce the dimension of the strategy space. For example, people may not be able to remember exactly what a certain player chose in the last ten moves, but may probably have a rough sense of how often did he/she cooperate. What if agents with higher memory capacity use heuristic strategies (e.g., if the opponent often cooperate, I also cooperate)?

3. It would be great if the authors could suggest some mechanisms that can potentially resolve the memory dilemma.

4. Researchers in other fields including psychology and economics have also investigated the relationship between cognitive/memory capacity and the results of social interactions (e.g., Duffy & Smith, 2014). Discussions on the relationship between these studies and the current one in the introduction or discussion part may attract a wider audience for this paper.

Reference:

Duffy, S., & Smith, J. (2014). Cognitive load in the multi-player prisoner's dilemma game: Are there brains in games? Journal of Behavioral and Experimental Economics, 51, 47–56. http://doi.org/10.1016/j.socec.2014.01.006

Reviewer #2: Please see my comment in the attachment!

Reviewer #3: This paper explores competition between different strategy spaces in the infinitely repeated donation game. The competition occurs between strategy spaces requiring different memory capacities, namely unconditional strategies (U), reactive strategies (R) and memory-1 strategies (M). Due to the asymmetry induced by competition between strategy spaces, the authors assume that behavioral update occurs via introspection dynamics throughout the course of the repeated game, rather than through e.g. imitation dynamics. They find that there is a tradeoff between the quality of strategies available to a player with a given memory capacity and the ease with which high quality strategies can be found via introspection dynamics. They show that, depending on the strength of selection and the acuteness of the social dilemma encoded by the donation game, this tradeoff can lead to either U, R or M dominating. Most interestingly, they also explore replicator dynamics at level of the supergame and identify a “memory-dilemma” in which R evolves even though it does not maximize population-level payoff from cooperation.

In some ways the tradeoff at the heart of this model is a familiar one, which crops up frequently in evolutionary dynamics as a tension between robustness and evolvability. However that familiarity is not to the determinant of this work, and indeed the case the authors explore is especially fascinating due to the way it results in a social dilemma reproducing itself at different scales (i.e. the donation game formulation of the Prisoner’s Dilemma becomes a Memory Dilemma in the supergame). I therefore believe that this paper will be of great interest to the wide readership of PLOS Computational Biology.

My only significant concern with the current manuscript is with the assumption that behavioral update occurs via introspection. This is a fine model to explore, but the central question here invites the reader to consider the implications for the evolution of memory, as the authors acknowledge in the discussion. It’s not at all clear that introspection dynamics apply to e.g. competition between desert spiders (line 323). The reason the authors focus on introspection dynamics is (line 132-137) that a player with a lower memory capacity cannot imitate the strategy of a player with a higher memory capacity. However there is a very natural way that such imitation could happen: a U player could imitate an M player by “imitating” the average rate of cooperation of the M player. An R player could imitate an M player by imitating the average observed rate of cooperation of an M player in response to cooperation by an opponent, and so on. It seems likely that the reduced ability to accurately imitate induced by low memory capacity may undermine some of the results presented here, and I would be very interested for the authors to explore this.

**Have the authors made all data and (if applicable) computational code underlying the findings in their manuscript fully available?**

Reviewer #1: None

Reviewer #2: **No: **The raw data is only available upon request.

Reviewer #3: Yes

PLOS authors have the option to publish the peer review history of their article (what does this mean?). If published, this will include your full peer review and any attached files.

Reviewer #1: No

Reviewer #2: No

Reviewer #3: **Yes: **Alexander J. Stewart
---

## [Decision Letter · Decision Letter 1]

28 Apr 2022

Dear Ms Schmid,

We are pleased to inform you that your manuscript 'Direct reciprocity between individuals that use different strategy spaces' has been provisionally accepted for publication in PLOS Computational Biology.

Best regards,

Lusha Zhu, Ph.D.

Associate Editor

PLOS Computational Biology

James O'Dwyer

Deputy Editor

PLOS Computational Biology

Reviewer's Responses to Questions

**Comments to the Authors:**

Reviewer #1: The authors have addressed all of my concerns, and I have no further issues. I appreciate the additional analyses done by the authors, as well as their thorough and detailed reply to my questions.

Reviewer #2: I am glad that the authors made significant progresses for this manuscript. I am happy to sign off at the moment as most of my concerns have been addressed nicely!

Reviewer #3: The authors have substantially revised the manuscript and thoroughly addressed my comments, and so I gladly recommend the paper be accepted.

**Have the authors made all data and (if applicable) computational code underlying the findings in their manuscript fully available?**

Reviewer #1: Yes

Reviewer #2: Yes

Reviewer #3: Yes

PLOS authors have the option to publish the peer review history of their article (what does this mean?). If published, this will include your full peer review and any attached files.

Reviewer #1: No

Reviewer #2: No

Reviewer #3: **Yes: **Alexander J Stewart

---

## [Editor Report · Acceptance letter]

24 May 2022

PCOMPBIOL-D-21-02127R1 

Direct reciprocity between individuals that use different strategy spaces

Dear Dr Schmid,

I am pleased to inform you that your manuscript has been formally accepted for publication in PLOS Computational Biology. Your manuscript is now with our production department and you will be notified of the publication date in due course.

With kind regards,

Agnes Pap
